# A Unified View on Solving Objective Mismatch in Model-Based Reinforcement Learning

**Ran Wei**\*                                                                                   *rw422@tamu.edu*
*VERSES Research Lab*

**Nathan Lambert**                                                                        *nathanl@allenai.org*
*Allen Institute for AI*

**Anthony McDonald**                                                                  *admcdonald@wisc.edu*
*University of Wisconsin-Madison*

**Alfredo Garcia**                                                                     *alfredo.garcia@tamu.edu*
*Texas A&M University*

**Roberto Calandra**                                                        *roberto.calandra@tu-dresden.de*
*TU Dresden*

**Reviewed on OpenReview:** *https://openreview.net/forum?id=tQVZguXhZb*

## Abstract

Model-based Reinforcement Learning (MBRL) aims to make agents more sample-efficient, adaptive, and explainable by learning an explicit model of the environment. While the capabilities of MBRL agents have significantly improved in recent years, how to best learn the model is still an unresolved question. The majority of MBRL algorithms aim at training the model to make accurate predictions about the environment and subsequently using the model to determine the most rewarding actions. However, recent research has shown that model predictive accuracy is often not correlated with action quality, tracing the root cause to the *objective mismatch* between accurate dynamics model learning and policy optimization of rewards. A number of interrelated solution categories to the objective mismatch problem have emerged as MBRL continues to mature as a research area. In this work, we provide an in-depth survey of these solution categories and propose a taxonomy to foster future research.

## 1 Introduction

Reinforcement learning (RL) has demonstrated itself as a promising tool for complex optimization landscapes and the creation of artificial agents by exceeding human performance in games (Mnih et al., 2015; Silver et al., 2016), discovering computer algorithms (Fawzi et al., 2022), managing power plants (Degrave et al., 2022), and numerous other tasks. The premise of RL is that complex agent behavior and decisions are driven by a desire to maximize cumulative rewards in dynamic environments (Silver et al., 2021). RL methods focus on learning a reward-optimal policy from sequences of state-action-reward tuples. These methods can be broadly classified as model-free RL (MFRL) and model-based RL (MBRL). MFRL methods directly learn the policy from the environment samples, whereas MBRL approaches learn an explicit model of the environment and use the model in the policy-learning process. MBRL methods are advantageous because they can make deep RL agents more sample-efficient, adaptive, and explainable. Prior work has shown that MBRL methods allow agents to plan with respect to variable goals or diverse environments (Zhang et al.,

---

\*Work done at Texas A&M University

2018; Hafner et al., 2023) and that designers can introspect an agent's decisions (van der Waa et al., 2018), which can help in identifying potential causes for failures (Räuker et al., 2023).

Despite the benefits of MBRL, there is considerable divergence in existing algorithms and no consensus on the aspects of the environment to model and how the model should be learned (e.g., model architecture, data arrangement, objective functions). For example, Dyna-style MBRL algorithms are trained to make accurate predictions about the environment, then find the optimal actions or policy with respect to the trained model (Sutton & Barto, 2018). The intuition behind these approaches is that improving the model's accuracy in predicting environment dynamics should facilitate better action selection and improved algorithm performance. However, recent research found that improved model accuracy often does not correlate with higher achieved returns (Lambert et al., 2020). While the underperformance of policies trained on the learned models is often due to the models' inability to sufficiently capture environment dynamics and the policies exploiting errors in the models (Jafferjee et al., 2020), Lambert et al. (2020) attributed the root cause of this phenomenon to the *objective mismatch* between model learning and policy optimization: while the policy is trained to maximize return, the model is trained for a different objective and not aware of its role in the policy decision-making process. This objective mismatch problem represents a substantial and fundamental limitation of MBRL algorithms, and resolving it will likely lead to enhanced agent capabilities.

In this review, we study existing literature and provide a unifying view of different solutions to the objective mismatch problem. Our main contribution is a taxonomy of four categories of decision-aware MBRL approaches: *Distribution Correction, Control-As-Inference, Value-Equivalence, and Differentiable Planning*, which derive modifications to the model learning, policy optimization, or both processes for the purpose of aligning model and policy objectives and gaining better performance (e.g., achieving higher returns). For each approach, we discuss its intuition, implementation, and evaluations, as well as implications for agent behavior and applications. This review is complementary to prior broader introductions of MBRL (e.g., see Moerland et al. (2023); Luo et al. (2022)) in its in-depth analysis of solutions to the objective mismatch problem and illustrations of implications for MBRL approaches.

## 2    Background

To facilitate comparisons between the reviewed approaches, we adopt the common notation and premise for MBRL based on (Sutton & Barto, 2018). In the subsequent sections, we introduce Markov Decision Processes, reinforcement learning, and the objective mismatch problem.

### 2.1    Markov Decision Process

We consider reinforcement learning in Markov Decision Processes (MDP) defined by tuple $(\mathcal{S}, \mathcal{A}, M, R, \mu, \gamma)$, where $\mathcal{S}$ is the set of states, $\mathcal{A}$ is the set of actions, $M : \mathcal{S} \times \mathcal{A} \to \Delta(\mathcal{S})$ is the environment transition probability function (also known as the dynamics model), $R : \mathcal{S} \times \mathcal{A} \to \mathbb{R}$ is the reward function, $\mu : \mathcal{S} \to \Delta(\mathcal{S})$ is the initial state distribution, and $\gamma \in [0, 1)$ is the discount factor. The RL agent interacts with the environment using a policy $\pi : \mathcal{S} \to \Delta(\mathcal{A})$ and generates trajectories $\tau = (s_{0:T}, a_{0:T})$ distributed according to $P(\tau)$ and evaluated by discounted cumulative rewards (also known as the return) $R(\tau)$. $P(\tau)$ and $R(\tau)$ are defined respectively as:

$$P(\tau) = \mu(s_0) \prod_{t=0}^{T} M(s_{t+1}|s_t, a_t)\pi(a_t|s_t), \quad R(\tau) = \sum_{t=0}^{T} \gamma^t R(s_t, a_t). \tag{1}$$

Abusing notation, we also use $P(\tau)$ and $R(\tau)$ to refer respectively to the probability measure and discounted reward for infinite horizon sequences $\tau = (s_{0:\infty}, a_{0:\infty})$. We further denote the marginal state-action density of policy $\pi$ in the environment $M$ (also known as the normalized occupancy measure) as $d_M^\pi(s, a) = (1 - \gamma)\mathbb{E}_{P(\tau)}[\sum_{t=0}^{\infty} \gamma^t \Pr(s_t = s, a_t = a)]$.

---

**Algorithm 1** Basic algorithm of model-based reinforcement learning

---

**Require:** Environment, dynamics model $\hat{M}$, policy $\pi$, data buffer $\mathcal{D}$, time budget $T$
  **while** $t \leq T$ **do**
    Interact with the environment and collect data $\mathcal{D} \leftarrow \mathcal{D} \cup (s, a, r, s')$
    Update model $\hat{M}$
    Update policy $\pi$
  **end while**

---

## 2.2 Reinforcement Learning

The goal of the RL agent is to find a policy that maximizes the expected return $J_M(\pi)$ in the environment with dynamics $M$, where $J_M(\pi)$ is defined as:

$$J_M(\pi) = \mathbb{E}_{P(\tau)}[R(\tau)]. \tag{2}$$

Importantly, the agent does not know the true environment dynamics and has to solve (2) without this knowledge. We assume the reward function is known.

(2) is often solved by estimating the state value function $V(s)$ and state-action value function $Q(s,a)$ of the optimal policy using the Bellman equations:

$$Q(s,a) = R(s,a) + \gamma \mathbb{E}_{s' \sim P(\cdot|s,a)}[V(s')],$$
$$V(s) = \max_{a \in \mathcal{A}} Q(s,a). \tag{3}$$

Then the optimal policy can be constructed by taking actions according to:

$$\pi(a|s) \in \arg\max_{a \in \mathcal{A}} Q(s,a). \tag{4}$$

Variations to (3) and (4) exist depending on the problem at hand. For example, in continuous action space, the maximum over action is typically found approximately using gradient descent (Lillicrap et al., 2015; Schulman et al., 2017a). Policy iteration algorithms estimate the value of the current policy by defining $V(s) = \mathbb{E}_{a \sim \pi(\cdot|s)}[Q(s,a)]$ and improve upon the current policy. We refer to the process of finding the optimal policy for an MDP, including these variations, as *policy optimization*.

In model-free RL, the expectation in (3) is estimated using samples from the environment. Model-based RL instead learns a model $\hat{M}$ and estimates the value function using samples from the model (often combined with environment samples). These algorithms alternate between data collection in the environment, updating the model, and improving the policy (see Algorithm 1). We will be mostly concerned with forward dynamics model of the form $\hat{M}(s'|s,a)$, although other types of dynamics models such as inverse dynamics models, can also be used (Chelu et al., 2020).

We use $Q_M^\pi(s,a)$ and $V_M^\pi(s)$ to distinguish value functions associated with different policies and dynamics. When it is clear from context, we drop $M$ and $\pi$ to refer to value functions of the optimal policy with respect to the *learned* dynamics model, since all reviewed methods are model-based. We treat all value functions as estimates since the true value functions can not be obtained directly.

## 2.3 Model Learning and Objective Mismatch

Many MBRL algorithms train the model $\hat{M}$ to make accurate predictions of environment transitions. This is usually done via maximum likelihood estimation (MLE):

$$\max_{\hat{M}} \mathbb{E}_{(s,a,s') \sim \mathcal{D}} \left[ \log \hat{M}(s'|s,a) \right]. \tag{MLE 5}$$

This choice is justified by the well-known simulation lemma (Kearns & Singh, 2002), which has been further refined in recent state-of-the-art MBRL algorithms to account for off-policy learning:

**Theorem 2.1.** *(Lemma 3 in (Xu et al., 2020)) Given an MDP with bounded reward:* $\max_{s,a} |R(s,a)| = R_{max}$ *and dynamics* $M$, *a data-collecting behavior policy* $\pi_b$, *and a learned model* $\hat{M}$ *with* $\mathbb{E}_{(s,a)\sim d_M^{\pi_b}} D_{KL}[M(\cdot|s,a)||\hat{M}(\cdot|s,a)] \leq \epsilon_{\hat{M}}$, *for arbitrary policy* $\pi$ *with bounded divergence* $\epsilon_\pi \geq \max_s D_{KL}[\pi(\cdot|s)||\pi_b(\cdot|s)]$, *the policy evaluation error is bounded by:*

$$|J_{\hat{M}}(\pi) - J_M(\pi)| \leq \frac{\sqrt{2}\gamma R_{\max}}{(1-\gamma)^2}\sqrt{\epsilon_{\hat{M}}} + \frac{2\sqrt{2}\gamma R_{\max}}{(1-\gamma)^2}\sqrt{\epsilon_\pi}. \tag{6}$$

Thus, one way to reduce the policy evaluation error of the optimizing policy $\pi$ is to make the model as accurate as possible in state-action space visited by the behavior policy while maintaining small policy divergence. There are two issues with this approach. First, unlike the tabular setting, the dynamics error might not reduce to zero even with infinite data in complex, high dimensional environments due to limited model capacity or model misspecification. Second, maintaining small policy divergence in the second term requires knowledge of the behavior policy in all states and additional techniques for constrained policy optimization. The former requirement can be demanding with an evolving behavior policy as in most online RL settings, or with an unknown behavior policy as in most offline RL settings, and the latter requirement can be undesirable since the goal in most RL settings is to quickly improve upon the behavior policy.

Towards better understanding of model learning and policy performance, Lambert et al. (2020) found that model predictive accuracy is often not correlated with the achieved returns. They attributed the root cause of this finding to the fact that, in the common practice, the model learning objective (i.e., maximizing likelihood) is different from the policy optimization objective (i.e., maximizing return) and they coined the term "objective mismatch" to refer to this phenomenon. One of the main manifestations of objective mismatch is that the dynamics model learned by one-step prediction can be inaccurate for long-horizon rollouts due to compounding error (Lambert et al., 2022). These inaccuracies can then be exploited during policy optimization (Jafferjee et al., 2020). Various methods have been proposed to improve the dynamics model's long-horizon prediction performance or avoid exploitation by the policy, for example by using multistep objectives (Luo et al., 2018), training the model to self-correct (Talvitie, 2017), directly predicting the marginal density (Janner et al., 2020), training a model with different temporal abstractions (Lambert et al., 2021), or quantifying epistemic uncertainty in the model (Chua et al., 2018).

Nevertheless, the relationship between model learning and policy optimization has been found to be highly nuanced. There also exist studies highlighting the diminishing contribution of model accuracy to policy performance (Palenicek et al., 2023). From a Bayesian RL perspective, model learning from one-step transitions is theoretically optimal as it provides the sufficient statistics for both model learning and policy optimization (Duff, 2002; Ghavamzadeh et al., 2015). However, the optimal Bayesian RL policy has to account for the uncertainty in the dynamics model parameters and maintaining such uncertainty is generally intractable. Thus, the complex interaction between model learning and policy optimization and efficient methods to bridge these misaligned objectives remains a substantial research gap.

## 2.4 Towards Decision-Aware MBRL

Overcoming the objective mismatch problem has important implications for safe and data-efficient RL. In domains where environment interaction is expensive or unsafe, off-policy or offline RL are used to extract optimal policies from a limited dataset (Levine et al., 2020). In offline MBRL, the dynamics model is typically fixed after an initial pretraining stage and other methods are required to prevent model-exploitation from the policy, such as by designing pessimistic penalties (Yu et al., 2020b; Kidambi et al., 2020). Decision-aware MBRL has the potential to simplify or remove the design of these post hoc methods.

Beyond data-efficiency and safety, decision-aware MBRL can potentially address the gaps in current automated decision-making software systems in various domains, such as transportation and health care (McAllister et al., 2022; Wang et al., 2021). These systems are traditionally developed and tested in a modular fashion. For example, in automated vehicles, the trajectory forecaster is typically developed independently of the vehicle controller. As a result, modules which pass the unit test may still fail the integration test.

Thus, the core motivation for solving the objective mismatch problem is to improve MBRL agent capabilities and downstream performance by designing model learning objectives that are aware of its role in or directly contribute to policy optimization, policy objectives that are aware of model deficiencies, or unified objectives that contribute to both (Lambert et al., 2020). It is therefore desirable if guiding principles can be identified to facilitate the design of future objectives. The goal of this survey is to identify these principles by synthesizing existing literature.

## 3  Survey Scope and Related Work

The focus of this survey is on existing approaches that address the objective mismatch problem in MBRL. We identified these approaches by conducting a literature search of the terms "objective mismatch," "model-based," and "reinforcement learning" and their permutations on Google Scholar and Web of Science. We compounded these searches by examining articles citing (Lambert et al., 2020) and their reference lists. We focused on (Lambert et al., 2020) because it was the first work to coin the term "objective mismatch" and the reference tree rooted at this paper led us to valuable older works that did not used this term. The initial search yielded 85 results which were screened for relevance. To be included in the survey, an article has to specifically address a solution to the objective mismatch problem or propose decision-aware objectives for model learning and policy optimization. Articles that simply discussed the objective mismatch problem without providing a solution were excluded. After abstract and full text screening, we retained a total of 46 papers [1].

Before presenting the included papers, we briefly cover related research areas that are deemed out-of-scope by our screening criteria in a non-exhaustive manner; these include state abstraction, representation learning, control-aware dynamics learning, decision-focused learning, and meta reinforcement learning.

MDP state abstraction focuses on aggregating raw state features while preserving relevant properties of the dynamics model, policy, or value function (Li et al., 2006; Abel et al., 2020). For example, bisimulation-based aggregation methods aim to discover models with equivalent transitions and have been applied in the deep RL setting (Zhang et al., 2020). Recent works on representation learning for control aim to factorize controllable representations from uncontrollable latent variables using, for example, inverse dynamics modeling (Lamb et al., 2022). These research directions are related to the objective mismatch problem in that they also aim to overcome the difficulty of modeling complex and irrelevant observations for control, however, they are out-of-scope for the current survey because they do not consider the mutual adaptation of model and policy while learning. Control-aware dynamics learning focus on dynamics model regularization based on smoothness or stabilizability principles in order to address compounding error in synthesizing classic (e.g., LQR) or learned controllers (Levine et al., 2019; Singh et al., 2021; Richards et al., 2023). However, they do not directly focus on the source of the problem, i.e., misaligned model and policy objectives, and thus are out-of-scope for the current survey. Objective mismatch is also related to the broad topic on decision-focused learning, where a model is trained to predict the parameters of an optimization problem, which is subsequently solved (Wilder et al., 2019; Elmachtoub & Grigas, 2022). Due to the focus on sequential decision-making problems (i.e., RL), more general work on decision-focused learning is out-of-scope.

Finally, we omit meta RL (Beck et al., 2023) in the survey and the related topics of Bayesian RL (Ghavamzadeh et al., 2015) and active inference, a Bayesian MBRL approach rooted in neuroscience (Da Costa et al., 2020; 2023). Meta RL based on the hidden parameter and partially observable MDP formulations (Doshi-Velez & Konidaris, 2016; Duff, 2002) has a particularly strong connection with decision-aware MBRL in that the policy is aware of the error, or rather uncertainty, in the dynamics model parameters and predictions by virtue of planning in the belief or hyper-state space. However, these approaches often focus on obtaining Bayes-optimal exploration strategies using standard or approximate Bayesian learning and belief-space planning algorithms as opposed to developing novel model and policy objectives (Zintgraf et al., 2019). We also side-step the open question on unified objectives in active inference (Millidge et al., 2021; Imohiosen et al., 2020) but discuss potential connections between active inference and control-as-inference (Levine, 2018; Millidge et al., 2020) in section 4.2.

---

[1]The full list of papers can be found at `https://github.com/ran-weii/objective_mismatch_papers`.

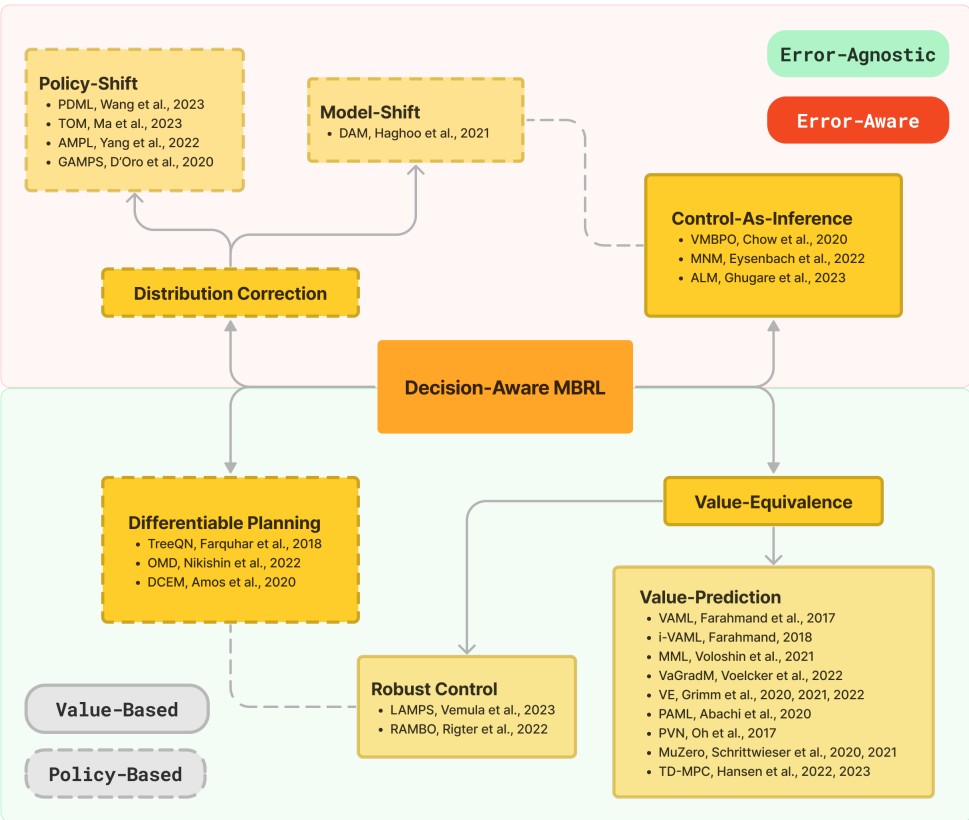

Figure 1: Schematic of the relationships between the core surveyed decision-aware MBRL approaches. Direct relationships are shown in solid arrows. Indirect relationships are shown in dashed connections. The algorithms in each category are sorted by the order in which they are presented in the paper.

## 4  Taxonomy

Our synthesis of the 46 papers identified 4 broad categories of approaches to solving the objective mismatch problem. We consolidated these into the following taxonomy of decision-aware MBRL:

- **Distribution Correction** adjusts for the mismatched training data in both model learning and policy optimization.

- **Control-As-Inference** provides guidance for the design of model learning and policy optimization objectives by formulating both under a single probabilistic inference problem.

- **Value-Equivalence** searches for models that are equivalent to the true environment dynamics in terms of value function estimation.

- **Differentiable Planning** embeds the model-based policy optimization process in a differentiable program such that both the policy and the model can be optimized towards the same objective.

A schematic of the relationships between these approaches is shown in Figure 1. The figure illustrates that Value-Equivalence approaches are the most prevalent in the literature, however, more recent approaches have concentrated on distribution correction and control-as-inference. The remaining sections discuss these approaches with a focus on their model learning and policy optimization objectives. Comparisons of the design and evaluations of the core reviewed algorithms are provided in Table 1 and Table 2, respectively.

### 4.1 Distribution Correction

Distribution correction aims to correct for training errors attributable to policy optimization on samples not from the true environment (model-shift) or to samples from an outdated or different policy (policy-shift). Our literature search identified one approach focused on addressing model-shift and four approaches focused on policy-shift.

#### 4.1.1 Addressing Model-Shift

Haghgoo et al. (2021) proposed to address model-shift by using the learned model as the proposal distribution for an importance sampling estimator of the expected return in the true environment as defined in (7):

$$\mathbb{E}_{P(\tau)}[R(\tau)] = \mathbb{E}_{Q(\tau)}\left[\frac{P(\tau)}{Q(\tau)}R(\tau)\right] = \mathbb{E}_{Q(\tau)}[w(\tau)R(\tau)].\tag{7}$$

where $P(\tau)$ is the trajectory distribution in the true environment, $Q(\tau) = \mu(s_0)\prod_{t=0}^{\infty}\hat{M}(s_{t+1}|s_t, a_t)\pi(a_t|s_t)$ is the trajectory distribution induced by the learned model $\hat{M}$ (known as the proposal distribution in importance sampling), and $w(\tau)$ is the importance weight. Since importance sampling is agnostic to the proposal distribution (i.e., $Q(\tau)$), this method can be applied to models trained using any objectives.

Given that both $P(\tau)$ and $Q(\tau)$ correspond to the MDP structure, the importance weight can be decomposed over time steps as:

$$w(s_{0:t}, a_{0:t}) = \prod_{m=0}^{t-1}\frac{M(s_{m+1}|s_m, a_m)}{\hat{M}(s_{m+1}|s_m, a_m)} = \prod_{m=0}^{t-1}w(s_m, a_m, s_{m+1}).\tag{8}$$

Although not directly available, the per-time step importance weight $w(s_m, a_m, s_{m+1})$ can be obtained using density ratio estimation through binary classification, similar to Generative Adversarial Networks (Sugiyama et al., 2012; Goodfellow et al., 2020). The policy is then optimized with respect to the importance-weighted reward. Thus, the authors refer to this method as Discriminator Augmented MBRL (DAM).

To address the possibility that the estimator can have high variance for long trajectories due to multiplying the importance weights, the authors proposed to optimize towards a model that achieves the lowest variance for the estimator, which is shown in (Goodfellow et al., 2016) to have the following form:

$$Q^*(\tau) \propto P(\tau)R(\tau).\tag{9}$$

Such a model can be found by minimizing $D_{KL}[Q^*(\tau)||Q(\tau)]$, resulting in a return-weighted model training objective. DAM's model learning and policy optimization objectives are thus defined as follows:

$$\max_{\hat{M}} \mathbb{E}_{P(\tau)}\left[\sum_{t=0}^{\infty}R(\tau)\log\hat{M}(s_{t+1}|s_t, a_t)\right],$$

$$\max_{\pi} \mathbb{E}_{Q(\tau)}\left[\sum_{t=0}^{\infty}\gamma^t\left(\prod_{m=0}^{t}w(s_m, a_m, s_{m+1})\right)R(s_t, a_t)\right].\tag{DAM 10}$$

The authors showed that DAM outperforms the standard Dyna MBRL algorithm with a MLE model objective in custom driving environments with multi-modal environment dynamics. They further found that including the importance weights in the policy objective is crucial for its superior performance.

#### 4.1.2 Addressing Policy-Shift

Rather than focusing on model-shift, the policy-shift approaches address objective mismatch by re-weighting the model training data such that samples less relevant or collected far away from the current policy's marginal state-action distribution are down-weighted.

Model training data in MBRL is typically collected by policies different from the current policy due to continuous policy updating. Wang et al. (2023) found that when training the model to make accurate

predictions on all data, the model produces more error on the most recent data compared to its overall performance. To this end, they propose an algorithm called Policy-adapted Dynamics Model Learning (PDML) which stores all historical policies and uses an estimate of the divergence between the data-collecting policy and the current policy to weight data samples for MLE model training.

In line with Wang et al. (2023) to address inferior model predictions on updated policies, Ma et al. (2023) proposed a weighted model training scheme motivated by the following lower bound of the log-transformed expected return:

$$
\begin{aligned}
\log J_M(\pi) &= \log \mathbb{E}_{d_M^\pi(s,a,s')}[R(s,a)] \\
&= \log \mathbb{E}_{d_{\hat{M}}^\pi(s,a,s')} \left[ \frac{d_M^\pi(s,a,s')}{d_{\hat{M}}^\pi(s,a,s')} R(s,a) \right] \\
&\geq \mathbb{E}_{d_{\hat{M}}^\pi(s,a,s')} \left[ \log \frac{d_M^\pi(s,a,s')}{d_{\hat{M}}^\pi(s,a,s')} + \log R(s,a) \right] \\
&\geq -D_f[d_{\hat{M}}^\pi(s,a,s')||d_M^\pi(s,a,s')] + \mathbb{E}_{d_{\hat{M}}^\pi(s,a,s')}[\log R(s,a)] ,
\end{aligned}
\tag{11}
$$

where $d_M^\pi(s,a,s') = (1-\gamma)\mathbb{E}[\sum_{t=0}^\infty \gamma^t \Pr(s_t = s, a_t = a, s_{t+1} = s')]$ denotes the normalized occupancy measure for transitions $(s,a,s')$, and $D_f$ denotes a chosen $f$-divergence measure. The first term in the lower bound suggests that the learned dynamics model should induce similar transition occupancy to the true environment under the current policy. To this end, the authors proposed the Transition Occupancy Matching (TOM) algorithm to optimize the first term while alternating with optimizing the second term using model-based policy optimization (MBPO) (Janner et al., 2019).

Although one could optimize the first term by simulating $\pi$ in $\hat{M}$, such an approach requires repeated simulation for each dynamics model update and fails to leverage data collected by the behavior policy. Instead, the authors proposed using dual RL techniques to minimize $D_f$ with respect to $d_{\hat{M}}^\pi$ where a dual value function $\tilde{Q}(s,a)$ is introduced to penalize $d_{\hat{M}}^\pi$ from violating the Bellman flow constraint (Nachum & Dai, 2020). Upon obtaining the optimal dual value function $\tilde{Q}^*(s,a)$, which can be estimated from the collected data, the dual RL formulation yields the following importance weight estimator for finding the optimal transition occupancy-matching dynamics using weighted MLE:

$$
w_{\text{TOM}}(s,a,s') = \frac{d_{\hat{M}}^\pi(s,a,s')}{d_M^{\pi_b}(s,a,s')} = f'_\star \left( \log \frac{d_M^\pi(s,a,s')}{d_M^{\pi_b}(s,a,s')} + \gamma \mathbb{E}_{\pi(a'|s')}[\tilde{Q}^*(s',a')] - \tilde{Q}^*(s,a) \right) ,
\tag{12}
$$

where the log density ratio on the right hand side is estimated using a discriminator and $f'_\star$ is the derivative of the Fenchel conjugate of the chosen $f$-divergence function. The resulting TOM model and policy objectives are as follows:

$$
\begin{aligned}
&\max_{\hat{M}} \; \mathbb{E}_{(s,a,s')\sim\mathcal{D}} \left[ w_{\text{TOM}}(s,a,s') \log \hat{M}(s'|s,a) \right] , \\
&\max_{\pi} \; \mathbb{E}_{(s,a)\sim d_{\hat{M}}^\pi} [\log R(s,a)] .
\end{aligned}
\tag{TOM 13}
$$

Empirically, the authors showed that TOM correctly assigns higher importance weights $w$ to transition samples more similar to the current policy and that TOM is more sample efficient and achieves higher asymptotic performance in MuJoCo environments (Todorov et al., 2012) than MBPO and PDML which uses heuristic policy divergence based weighting. The authors also compared TOM with VaGradM (a different class of decision-aware MBRL algorithm reviewed in section 4.3.1) and found TOM performs significantly better except for one environment.

In contrast to the above online RL approaches, Yang et al. (2022) addressed the offline RL setting and proposed the Alternating Model and Policy Learning (AMPL) algorithm to optimize the following lower

bound of the expected return:

$$
\begin{aligned}
J_M^\pi &\geq J_{\hat{M}}^\pi - |J_M^\pi - J_{\hat{M}}^\pi| \\
&\geq J_{\hat{M}}^\pi - \frac{\gamma R_{max}}{\sqrt{2}(1-\gamma)}\sqrt{D_\pi(M, \hat{M})}\,,
\end{aligned}
\tag{14}
$$

where the divergence in the second term which characterizes policy-shift is defined as:

$$
D_\pi(M, \hat{M}) = \mathbb{E}_{(s,a)\sim d_M^{\pi_b}}\left[w_{\text{AMPL}}(s,a)D_{KL}[M(s'|s,a)\pi_b(a'|s')||\hat{M}(s'|s,a)\pi(a'|s')]\right]
\tag{15}
$$

and $w_{\text{AMPL}}(s,a) = \frac{d_M^\pi(s,a)}{d_M^{\pi_b}(s,a)}$ is the marginal state-action density ratio between the current policy and the behavior policy.

Similar to TOM, the bound in (14) suggests MLE dynamics model learning weighted by $w_{\text{AMPL}}(s,a)$. However, for policy training, (14) suggests not only maximizing reward but also minimizing the divergence so that the evaluation error in the second term is controlled for. Instead of using dual-RL techniques to estimate $w$ as in TOM, the authors proposed a novel fixed point estimator. They further approximated $D_\pi(M, \hat{M})$ for policy optimization using the output of a discriminator: $\log(1 - C(s,a))$. The APML model and policy objectives are thus defined as follows:

$$
\begin{aligned}
&\max_{\hat{M}} \mathbb{E}_{(s,a,s')\sim\mathcal{D}}[w_{\text{AMPL}}(s,a)\log\hat{M}(s'|s,a)]\,, \\
&\max_\pi \mathbb{E}_{(s,a)\sim d_{\hat{M}}^\pi}[R(s,a) - \log(1 - C(s,a))]\,.
\end{aligned}
\tag{AMPL 16}
$$

A similar method was proposed in (Hishinuma & Senda, 2021) which instead estimates the importance weight using full-trajectory model rollouts.

Empirically, Yang et al. (2022) showed that APML outperforms a number of SOTA model-based and model-free offline RL algorithms (e.g., MOPO; (Yu et al., 2020b), COMBO (Yu et al., 2021), CQL (Kumar et al., 2020)) in most D4RL environments (Fu et al., 2020) and that including the importance weight $w$ in the model objective is crucial for its performance.

Also in the offline RL setting, D'Oro et al. (2020) focused specifically on policy-gradient estimation using samples from the offline dataset, but the value function is estimated using the learned model (model-value gradient; MVG). This setup can be advantageous since the model bias is limited to the value estimate but not the sample-based evaluation of expectation. They proposed the Gradient-Aware Model-based Policy Search (GAMPS) algorithm which trains the model to minimize the following bound on the difference from the true policy-gradient:

$$
\|\nabla_\pi J_M(\pi) - \nabla_\pi J^{\text{MVG}}(\pi)\| \leq \frac{\gamma\sqrt{2}ZR_{max}}{(1-\gamma)^2}\sqrt{\mathbb{E}_{(s,a)\sim\eta_P^\pi}D_{KL}[M(\cdot|s,a)||\hat{M}(\cdot|s,a)]}\,,
\tag{17}
$$

where $Z$ is a constant and $\eta_P^\pi$ is a state-action density weighted by the policy-gradient norm. This suggests a model training objective where samples with higher policy-gradient norms are up-weighted.

The GAMPS model and policy objectives are defined as follows:

$$
\begin{aligned}
&\max_{\hat{M}} \mathbb{E}_{\tau\sim\mathcal{D}}\left[\sum_{t=0}^\infty \gamma^t\left(\prod_{m=0}^t \frac{\pi(a_t|s_t)}{\pi_b(a_t|s_t)}\sum_{l=0}^t \|\nabla\log\pi(a_l|s_l)\|\right)\log\hat{M}(s_{t+1}|s_t,a_t)\right]\,, \\
&\max_\pi \mathbb{E}_{\tau\sim\mathcal{D}}\left[\sum_{t=0}^\infty \gamma^t\left(\prod_{m=0}^t \frac{\pi(a_t|s_t)}{\pi_b(a_t|s_t)}\right)Q(s_t,a_t)\log\pi(a_t|s_t)\right]\,.
\end{aligned}
\tag{GAMPS 18}
$$

Empirically, the authors first used a diagnostic gridworld environment to show that GAMPS can accurately estimate policy gradients with a constrained dynamics model class which conditions only on actions but not on states. On two MuJoCo tasks, they showed that GAMPS achieved higher asymptotic performance than MLE model learning and two classical model-free RL algorithms adapted to the offline setting.

**Distribution Correction Summary**: All approaches in this category focused on return estimation under mismatched distributions. The single approach addressing *model-shift* adopted a straightforward importance sampling estimator, while the rest of the approaches have focused on bounding policy or policy-gradient evaluation error with respect to *policy-shift* as a result of deviating from the behavior policy. All approaches also adopted a weighted maximum likelihood model learning objective where the weights represent relevance to policy optimization, such as higher return, closeness to the current policy, or potential for policy improvement. However, by changing weights over the course of training, these models are able to adapt to changes in the current policy as opposed to being policy-agnostic as in standard MLE model learning.

## 4.2 Control-As-Inference

Besides the distribution correction approaches, other methods have attempted to leverage existing validated approaches to solve the objective mismatch problem. One such principled approach is to leverage the control-as-inference framework (Levine, 2018) and formulate both model learning and policy optimization as a single probabilistic inference problem.

The core concept of control-as-inference is that optimal control can be formulated as a probabilistic inference problem if we assume optimal behavior were to be observed in the future and compute a posterior distribution over the unknown actions that led to such behavior. Most control-as-inference methods define optimality using a binary variable $\mathcal{O}$ where the probability that an observed state-action pair is optimal ($\mathcal{O} = 1$) is defined as follows:

$$P(\mathcal{O}_t = 1|s_t, a_t) = \exp(R(s_t, a_t)). \tag{19}$$

The posterior, represented as a variational distribution $Q(\tau)$, is found by maximizing the lower bound of the marginal likelihood of optimal trajectories as follows:

$$
\begin{aligned}
\log P(\mathcal{O}_{0:\infty} = 1) &= \log \int_\tau P(\mathcal{O}_{0:\infty}, \tau) \\
&= \log \mathbb{E}_{P(\tau)}[P(\mathcal{O}_{0:\infty} = 1|\tau)] \\
&= \log \mathbb{E}_{P(\tau)}\left[\exp\left(\sum_{t=0}^{\infty} R(s_t, a_t)\right)\right] \\
&\geq \mathbb{E}_{Q(\tau)}\left[\sum_{t=0}^{\infty} R(s_t, a_t)\right] - D_{KL}[Q(\tau)||P(\tau)].
\end{aligned}
\tag{20}
$$

The prior $P(\tau)$ is usually defined using the structure of the MDP:

$$P(\tau) = \mu(s_0) \prod_{t=0}^{\infty} M(s_{t+1}|s_t, a_t) P(a_t|s_t), \tag{21}$$

where $P(a|s)$ is a prior or default policy usually set to a uniform distribution.

Most of the design decisions in control-as-inference are incorporated in defining the variational distribution $Q(\tau)$. Prior control-as-inference approaches, such as the ones reviewed in (Levine, 2018), define $Q(\tau)$ using the environment dynamics and do not incorporate a learned model in the formulation. Thus, the majority of those algorithms are model-free.

In contrast to those approaches, Chow et al. (2020) proposed a joint model learning and policy optimization algorithm under the control-as-inference framework called Variational Model-Based Policy Optimization (VMBPO) by incorporating the learned dynamics model in the variational distribution defined as follow:

$$Q(\tau) = \mu(s_0) \prod_{t=0}^{\infty} \hat{M}(s_{t+1}|s_t, a_t) \pi(a_t|s_t). \tag{22}$$

Although $Q(\tau)$ has the same structure as the proposal distribution in the importance sampling-based algorithm DAM (see (7)), the model learning and policy optimization objectives of VMBPO are derived from a single marginal likelihood lower bound defined as:

$$\max_{\hat{M},\pi} \; \mathbb{E}_{Q(\tau)}\left[\sum_{t=0}^{\infty}\gamma^t\left(R(s_t,a_t)+\log\frac{P(a_t|s_t)}{\pi(a_t|s_t)}+\log\frac{M(s_{t+1}|s_t,a_t)}{\hat{M}(s_{t+1}|s_t,a_t)}\right)\right]. \tag{VMBPO 23}$$

Defining a Bellman-like recursive equation based on (23):

$$Q(s,a)=R(s_t,a_t)+\log\frac{P(a|s)}{\pi(a|s)}+\mathbb{E}_{\hat{M}(s'|s,a)}\left[V(s')+\log\frac{M(s'|s,a)}{\hat{M}(s'|s,a)}\right]. \tag{24}$$

The authors showed that the optimal variational dynamics and policy have the following form:

$$\begin{aligned}
\hat{M}(s'|s,a) &\propto \exp(V(s')+\log M(s'|s,a)),\\
\pi(a|s) &\propto \exp(Q(s,a)+\log P(a|s)).
\end{aligned} \tag{25}$$

Similar to DAM, the authors used a discriminator to estimate the dynamics density ratio.

Interestingly, the optimal dynamics are encouraged to be not only similar to the true dynamics via the log likelihood term, but also have a higher tendency to sample high value states. Such an optimistic dynamics model is similar to that of DAM, albeit for a different reason that the dynamics is conditioned on the optimality variable while performing variational inference. Also similar to DAM's importance weight-augmented reward, the policy optimizes not only reward but also the negative log density ratio between the learned and the ground truth dynamics. This encourages the policy to visit states where the learned dynamics is accurate (lower cross entropy with the ground truth dynamics) and also states where the learned dynamics is uncertain (higher entropy).

Empirically, the authors showed that VMBPO outperforms SOTA model-based (e.g., MBPO) and model-free baselines (e.g., SAC (Haarnoja et al., 2018)) in six MuJoCo environments for a range of policy learning rate settings. However, they did not evaluate specific agent properties discussed in the previous paragraph.

While Chow et al. (2020) was the first to propose a unified decision-aware objective for model learning and policy optimization, Eysenbach et al. (2022) pointed out that the objective (23) is an upper-bound on the RL objective, which can be decomposed as the expected return and its variance, and thus undesirable. They instead proposed an algorithm called Mismatched-No-More (MNM) with an alternative optimality variable whose distribution is conditioned on trajectories instead of state-action pairs:

$$P(\mathcal{O}=1|\tau)=R(\tau)=\sum_{t=0}^{\infty}\gamma^t R(s_t,a_t). \tag{26}$$

Using this definition of the optimality variable, they derived a modified log marginal likelihood lower bound:

$$\begin{aligned}
\log P(\mathcal{O}=1) &= \log\int_{\tau}P(\mathcal{O}=1,\tau)\\
&= \log\mathbb{E}_{P(\tau)}[P(\mathcal{O}=1|\tau)]\\
&\geq \mathbb{E}_{Q(\tau)}[\log R(\tau)+\log P(\tau)-\log Q(\tau)]\\
&= \mathbb{E}_{Q(\tau)}\left[\log\sum_{t=0}^{\infty}\gamma^t R(s_t,a_t)+\log P(\tau)-\log Q(\tau)\right]\\
&\geq \mathbb{E}_{Q(\tau)}\left[\sum_{t=0}^{\infty}\gamma^t\log R(s_t,a_t)+\log P(\tau)-\log Q(\tau)\right].
\end{aligned} \tag{27}$$

The benefit of this modification is that it is a lower bound on the RL objective so that improving this bound guarantees improving upon agent performance.

MNM also optimizes a single objective for both model learning and policy optimization, except that the reward is log-transformed:

$$\max_{\hat{M},\pi} \ \mathbb{E}_{Q(\tau)} \left[ \sum_{t=0}^{\infty} \gamma^t \left( \log R(s_t, a_t) + \log \frac{P(a_t|s_t)}{\pi(a_t|s_t)} + \log \frac{M(s_{t+1}|s_t, a_t)}{\hat{M}(s_{t+1}|s_t, a_t)} \right) \right] . \tag{MNM 28}$$

To evaluate MNM, the authors first showed that in a gridworld goal-reaching environment, MNM solves the task faster than VMBPO and Q-learning and it is robust to constraints on the dynamics model to only make low-resolution predictions. On robotic control tasks with contact dynamics and sparse reward (e.g., door-opening) in MuJoCo, Deepmind Control suite (DMC) (Tassa et al., 2018), Metaworld (Yu et al., 2020a), and ROBEL (Ahn et al., 2020) environments, MNM frequently outperformed MBPO and SAC by a large margin and it consistently performed well in all tasks. Furthermore, the authors found that MNM's model-based value estimates were stable and did not explode towards large values throughout the learning process, which suggest robustness to model-exploitation. Visualizing the learned dynamics, the authors also found the MNM dynamics model tends to generate transitions towards high value states, which provides evidence for the optimistic dynamics to speed up learning.

Extending MNM to the visual RL setting, Ghugare et al. (2022) replaced the state dynamics model with a latent dynamics model $\hat{M}(z'|z, a)$ and an observation encoder $\hat{E}(z|s)$ in an algorithm called Latent Aligned Model (ALM). By designing the following prior and variational distributions:

$$P(\tau) = \mu(s_0) \prod_{t=0}^{\infty} M(s_{t+1}|s_t, a_t) P(a_t|z_t) \hat{E}(z_t|s_t) , \tag{29}$$

$$Q(\tau) = \mu(s_0) E(z_0|s_0) \prod_{t=0}^{\infty} M(s_{t+1}|s_t, a_t) \hat{M}(z_{t+1}|z_t, a_t) \pi(a_t|z_t) , \tag{30}$$

where $\pi(a|z)$ is a latent space policy, the ALM's model $\{\hat{M}, \hat{E}\}$ and policy jointly optimize the following objective:

$$\max_{\hat{M},\hat{E},\pi} \ \mathbb{E}_{Q(\tau)} \left[ \sum_{t=0}^{\infty} \gamma^t \left( \log R(s_t, a_t) + \log \frac{P(a_t|z_t)}{\pi(a_t|z_t)} + \log \frac{\hat{E}(z_{t+1}|s_{t+1})}{\hat{M}(z_{t+1}|z_t, a_t)} \right) \right] . \tag{ALM 31}$$

Interestingly, the third term in the augmented reward, which can be estimated using a discriminator similar to VMBPO and MNM, is the information gain of the latent dynamics model, thus connecting this approach with prior work on intrinsic motivation and information-theoretic approaches in RL (Eysenbach et al., 2021; Rakelly et al., 2021; Sun et al., 2011) and the pragmatic-epistemic value decomposition of the expected free energy objective function for planning in active inference (Millidge, 2020; Sajid et al., 2021; Fountas et al., 2020).

However, different from VMBPO and MNM's weighted-regression dynamics model learning objective, the ALM loss implies a "self-predictive" model learning process (Schwarzer et al., 2020), where the latent dynamics model is trained to predict the encoding of observations (and reward) in the collected dataset and the encoder is trained to match the predictions of the latent dynamics (similar to Bootstrap Your Own Latent (Grill et al., 2020)). While theoretical understanding of self-predictive objectives is still lacking, Subramanian et al. (2022) showed that, when reward prediction is also incorporated into the objective function, the learned latent state $z$ is a sufficient statistic (or information state) for the optimal policy with respect to the true environment and Tang et al. (2023) proposed that it implicitly performs eigen-decomposition of the true environment dynamics. Thus, the dynamics model learned by ALM is likely more task-agnostic than the dynamics models of VMBPO and MNM. This also suggests that decision-aware objectives may not necessarily need to bias model learning, aligning this method further with active inference.

Empirically, the authors found that ALM significantly outperformed SAC and SAC-SVG (Amos et al., 2021), a MLE-based MBRL algorithm with latent dynamics, with 2e5 environment steps (1/5 of the usual RL training steps) in five MuJoCo environments. Using the Q value evaluation protocol from (Chen et al.,

2021) and (Fujimoto et al., 2018), they found ALM value estimates to consistently have negative bias (underestimation). Similar to MNM, they also found that including the density ratio term in the objective is crucial.

> **Control-as-Inference Summary**: The control-as-inference category is closely related to the distribution-correction category in that the variational distribution can be interpreted as the proposal distribution for an importance sampling estimator of the expected return. However, control-as-inference focuses more directly on return optimization as opposed to estimation as signified by the maximization of the likelihood of optimal trajectories.
>
> The main advantage of the control-as-inference framework is that it provides a clear guidance to the derivation of model and policy objectives as long as a factorization of the prior and variational distributions are given. However, a major downside of current control-as-inference approaches is that the factorization of these distributions are largely hand-designed, which is the key to the attractive properties in (31) but potentially a bottleneck for future objective design. Although automated design of posterior distributions leveraging graphical model factorization has been proposed for variational inference in probabilistic predictive models (Webb et al., 2018), extensions to the RL setting have not been considered or developed.

### 4.3 Value-Equivalence

A core use case of the learned dynamics model is that the agent can sample from it to augment the limited environment samples for estimating the expected return or value function. Thus, as long as the dynamics can lead to accurate value estimates, it does not need to model the environment at all. In other words, the model is free to discover any dynamics that are equivalent to the true dynamics in terms of estimating value without wasting resources on irrelevant aspects of the environment. This class of approaches focuses almost entirely on model learning and performs standard model-based policy optimization. In this section, we first summarize how the literature formulates equivalent dynamics and how identifying such dynamics can be formulated as a value-prediction problem. We then summarize how such equivalence is related to the robustness properties of the learned dynamics.

#### 4.3.1 Value-Prediction

The value-prediction approaches propose to directly predict states with accurate values rather than accurate features. This is desirable if the dynamics model has limited capacity modeling all aspects of the environment faithfully, for example when using a uni-modal distribution to predict multi-model transitions, but the model can generate states whose values are close to the ground truth future state values. The models don't necessarily have to predict the true expected return value which is unknown, and a major design decision in these approaches is what value function to predict.

As the first approach in this class, Farahmand et al. (2017) proposed to train the dynamics model such that it yields the same Bellman backup (i.e., (3)) as the ground truth model. They formulated this intuition as the following objective called Value-Aware Model Loss (VAML):

$$\min_{\hat{M}} \ \mathbb{E}_{(s,a,s') \sim \mathcal{D}} \max_{V} \left| V(s') - \mathbb{E}_{s'' \sim \hat{M}(\cdot|s,a)}[V(s'')] \right|^2 . \qquad \text{(VAML 32)}$$

Since the true value function is not known, a robust formulation is used to optimize against the worst-case value function.

Empirically, the authors showed that VAML achieves smaller value-estimation error and higher return than MLE dynamics models in a finite MDP using a low-resolution dynamics model class.

To account for the fact that the optimizing policy $\pi$ may be different from the behavior policy $\pi_b$, Voloshin et al. (2021) introduced density ratio correction into an objective similar to VAML. Furthermore, to account for unknown density ratio and value function, they propose to optimize against the worst-case for both in

their Minimax Model Learning (MML) objective:

$$\min_{\hat{M}} \max_{w,V} \left| \mathbb{E}_{(s,a,s') \sim \mathcal{D}} \left[ w(s,a) \left( V(s') - \mathbb{E}_{s'' \sim \hat{M}(s'|s,a)}[V(s'')] \right) \right] \right| , \qquad \text{(MML 33)}$$

where $w(s,a) = \frac{d_M^\pi(s,a)}{d_M^{\pi_b}(s,a)}$ is the unknown density ratio between the optimizing policy and the behavior policy in the ground truth environment. They showed that this objective is a tighter bound on the policy evaluation error than the VAML objective.

Asadi et al. (2018) showed that the VAML loss function is equivalent to minimizing the Wasserstein distance between the learned and ground truth dynamics. As a result, the learned dynamics model tends to have attractive smoothness properties (e.g. being K-Lipschitz) and are less prone to compounding error and model exploitation.

However, the robust formulation of VAML poses a challenging optimization problem. As a response, Farahmand (2018) proposed to replace the worst-case value function with the most recent estimate in an objective called iterative VAML. In (Ayoub et al., 2020), the authors paired iterative VAML with an optimistic planning algorithm to derive a novel regret bound for the RL agent. Related to iterative VAML, MuZero (Schrittwieser et al., 2020; 2021) and Value Prediction Network (VPN) (Oh et al., 2017) replace the most recent value estimate with one bootstrapped from Monte Carlo Tree Search and multi-step look-ahead search, respectively (for a more nuanced discussion of VAML and MuZero, see (Voelcker et al., 2023)). These algorithms all train the model to perform reward-prediction in addition to value-prediction. A similar architecture and loss function was used in the Predictron model to evaluate the return of fixed deterministic policies (Silver et al., 2017a). MuZero has been shown to outperform prior state-of-the-art algorithms (e.g., AlphaZero; (Silver et al., 2017b)) in Go, Shogi, and Chess and large scale image-based online and offline RL settings (Schrittwieser et al., 2020; 2021).

More recently, Hansen et al. (2022; 2023) proposed combining value-prediction, reward-prediction, and self-prediction (i.e., the model learning objective in (31)) in their TD-MPC algorithm and performing model-predictive control with a learned terminal value function. They showed that with a small number of architectural adaptations to handle environments with distinct observation and action spaces and reward magnitudes, TD-MPC agents trained with a single set of hyperparameters substantially outperformed prior state of the art (e.g. Dreamer-v3 (Hafner et al., 2023)) on a large set of 104 continuous control tasks. Most notably, the authors showed in a multi-task setting across 80 tasks that increasing the number of model parameters led to substantial increase in returns and that multi-task training improved agent performance by almost two-fold when fine-tuned on new tasks compared to training from scratch.

Extending the VAML approach, Grimm et al. (2020) considered value-prediction with respect to *a set of* policies $\Pi$ and *a set of* arbitrary value functions $\mathcal{V}$ (i.e., not associated with any particular policies). They formulated the following objective based on the proposed Value-Equivalence (VE) principle:

$$\min_{\hat{M}} \sum_{\pi \in \Pi} \sum_{V \in \mathcal{V}} \mathbb{E}_{(s) \sim \mathcal{D}, a \sim \pi(\cdot|s)} \left\| \mathbb{E}_{s' \sim M(\cdot|s,a)}[V(s')] - \mathbb{E}_{s'' \sim \hat{M}(\cdot|s,a)}[V(s'')] \right\| . \qquad \text{(VE 34)}$$

The primary difference between the VE loss and the VAML loss is that it is formulated using arbitrary policies and values, rather than just the data-collecting policies and its associated value estimates. The benefit of this is that as we increase the size of the set of policies and value functions considered, the space of value-equivalent models shrink before eventually reducing to the single ground truth model. Furthermore, when the model has limited capacity even at predicting values, one can trade off between the policy and value set considered and the desired value-equivalence loss (Grimm et al., 2021; 2022).

Empirically, the authors showed that VE significantly outperformed MLE on two tabular environments (Catch and Four Rooms) and Cartpole when high rank constraints where imposed on the dynamics model, the agent performance improved with increasing size of the value function set.

Despite the simplicity and popularity of the VAML-family loss, Voelcker et al. (2022) suggested that it can cause undesirable optimization behavior when querying out-of-distribution value function estimates, especially when the learned model is not regularized to be close to the true environment. Using the inductive

bias that the learned model should predict states similar to the true environment for next state samples $s'$ in the dataset, they replaced the value function estimate in the VAML loss with its Taylor expansion around $s'$: $\hat{V}(s)|_{s'} = V(s') + (\nabla_s V(s)|_{s'})^\intercal (s - s')$. Assuming deterministic dynamics and applying the Cauchy Schwartz inequality: $(\sum_{i=1}^n x_i)^2 \leq n \sum_{i=1}^n x_i^2$, the VAML loss reduces to the following Value-Gradient Weighted Model Loss (VaGradM):

$$\min_{\hat{M}} \ \mathbb{E}_{(s,a,s')\sim\mathcal{D}, s''\sim\hat{M}(\cdot|s,a)} \left[ (s'' - s')^\intercal diag(\nabla_s V(s)|_{s'})(s'' - s') \right]. \tag{VaGradM 35}$$

When evaluated against VAML and MLE dynamics model training approaches with expressive dynamics (e.g., sufficiently large neural networks) in two MuJoCo environments, the authors found VaGradM to be robust to loss explosion when the value estimates were inaccurate in the initial training steps and converged to similar solutions to the MLE dynamics. However, VaGradM outperformed MLE when the dynamics model was constrained to fewer neural network layers and when distracting state dimensions were added.

The value-prediction approach can also be applied to policy-based RL methods, where instead of training the model to make accurate predictions of values, the model is trained to make accurate predictions of policy gradients. Abachi (2020) proposed the (multi-step) Policy-Aware Model Loss (PAML) defined as follows:

$$\min_{\hat{M}} \ \left\| \mathbb{E}_{\tau\sim P(\tau)} \left[ \sum_{k=0}^K \gamma^k F(s_t, a_t) \right] - \mathbb{E}_{\tau\sim Q(\tau)} \left[ \sum_{k=0}^K \gamma^k F(s_t, a_t) \right] \right\|_2, \tag{PAML 36}$$

where $Q(\tau) = \mu(s_0) \prod_{t=0}^\infty \hat{M}(s_{t+1}|s_t, a_t)\pi(a_t|s_t)$, $F(s,a) = \nabla \log \pi(a|s)Q^*(s,a)$, and $Q^*(s,a)$ is a model-free value estimate.

In a linear-quadratic control setting, the author found PAML to learn more slowly and achieves lower asymptotic performance than MLE but performs substantially better when irrelevant state dimensions are added. However, this result did not transfer to the MuJoCo environments where PAML and MLE only performed on par regardless of whether there were irrelevant dimensions.

> **Value-Prediction Summary**: Value-prediction approaches are similar to distribution correction and control-as-inference approaches in that it also tries to obtain more accurate value estimates, however, it focuses on estimating the Bellman operator as opposed to the return directly. Compared to two previous categories of approaches, value-prediction approaches have the benefit of requiring fewer moving parts, e.g., the importance weight estimator, and they do not by-default generate optimistic or pessimistic dynamics. However, by focusing on value-based RL algorithms, value-prediction approaches can be less flexible than the other classes of approaches.

### 4.3.2 Robust Control

While seemingly bearing no obvious connection to value-prediction, the robust control approaches reviewed in this section are in fact derived from the same value-equivalence principle for the purpose of learning value-equivalent dynamics to the ground truth environment dynamics. As we discuss below, these approaches highlight an inherent pessimism in value-equivalent dynamics models which is neglected in the value-prediction approaches.

Modhe et al. (2020; 2021) suggested that the VAML and value-equivalence loss functions can be understood from the perspective of model advantage defined as:

$$A_{\hat{M}}^\pi(s, s') = \gamma \left[ \mathbb{E}_{a\sim\pi(\cdot|s), s''\sim\hat{M}(\cdot|s,a)}[V(s'')] - V(s') \right], \tag{37}$$

where positive model advantage corresponds to optimistic dynamics and negative model advantage corresponds to pessimistic dynamics. However, by minimizing the norm of the value-prediction error, VAML and the VE optimize towards zero model advantage.

A more complete relationship between value-equivalence and model advantage is depicted in (Vemula et al., 2023) where the authors derived the following decomposition of the return gap between the optimizing policy $\pi$ and the behavior policy $\pi_b$ in terms of $\pi$'s value in the learned dynamics model:

$$(1-\gamma)[J_M(\pi) - J_M(\pi_b)] = \underbrace{\mathbb{E}_{(s,a)\sim d_M^{\pi_b}}\left[V_{\hat{M}}^\pi(s) - Q_{\hat{M}}^\pi(s,a)\right]}_{\text{Model-based advantage under data distribution}}$$

$$+ \gamma\underbrace{\mathbb{E}_{(s,a)\sim d_M^\pi}\left[\mathbb{E}_{s'\sim M(\cdot|s,a)}\left[V_{\hat{M}}^\pi(s')\right] - \mathbb{E}_{s''\sim\hat{M}(\cdot|s,a)}\left[V_{\hat{M}}^\pi(s'')\right]\right]}_{\text{Model dis-advantage under learner distribution}}$$

$$+ \gamma\underbrace{\mathbb{E}_{(s,a)\sim d_M^{\pi_b}}\left[\mathbb{E}_{s''\sim\hat{M}(\cdot|s,a)}\left[V_{\hat{M}}^\pi(s'')\right] - \mathbb{E}_{s'\sim M(\cdot|s,a)}\left[V_{\hat{M}}^\pi(s')\right]\right]}_{\text{Model advantage under data distribution}}. \tag{38}$$

The second and third terms in (38) suggest that the return gap between the two policies is more nuanced than just value-prediction error; it also involves model advantage and disadvantage under the data and learner distributions.

This relationship suggests a recipe for jointly optimizing model and policy to improve upon the behavior policy, namely, training the policy in the learned model starting from states visited by the behavior policy and simultaneously training the model to increase advantage under the data distribution and decrease advantage under the unknown learner distribution. Using this insight, Vemula et al. (2023) proposed an algorithm called Lazy Model-based Policy Search (LAMPS) with the following model and policy objectives:

$$\max_{\hat{M}} \mathbb{E}_{(s,a)\sim D,s'\sim\hat{M}(\cdot|s,a)}\left[V_{\hat{M}}^\pi(s')\right] - \mathbb{E}_{(s,a)\sim d_{\hat{M}}^\pi,s'\sim\hat{M}(\cdot|s,a)}\left[V_{\hat{M}}^\pi(s')\right],$$

$$\max_{\pi} \mathbb{E}_{s\sim D,a\sim\pi(\cdot|s)}[Q_{\hat{M}}^\pi(s,a)], \tag{LAMPS 39}$$

where $d_{\hat{M}}^\pi$ is obtained by simulating the learner policy in the learned dynamics. By optimizing (39), the dynamics model will be optimistic under the data distribution, similar to control-as-inference approaches. However, the dynamics model will be pessimistic rather than just entropic outside the data distribution, resulting in a robust formulation.

Empirically, the authors found LAMPS to consistently achieve higher performance with fewer environment steps than MBPO across four MuJoCo environments.

In the context of offline RL, Rigter et al. (2022) proposed a similar algorithm to LAMPS called Robust Adversarial Model-based Offline Policy Optimization (RAMBO). The RAMBO model and policy objectives are defined as follow:

$$\max_{\hat{M}} \lambda\mathbb{E}_{(s,a,s')\sim\mathcal{D}}[\log\hat{M}(s'|s,a)] - \mathbb{E}_{(s,a)\sim d_{\hat{M}}^\pi,s'\sim\hat{M}(\cdot|s,a)}\left[V_{\hat{M}}^\pi(s')\right],$$

$$\max_{\pi} \mathbb{E}_{(s,a)\sim d_{\hat{M}}^\pi}[R(s,a)], \tag{RAMBO 40}$$

where the model advantage loss (the first term in (39) model objective) is replaced with the standard MLE model objective and a hyperparameter $\lambda$ is used to weight against the second term optimizing model disadvantage. The policy is optimized using the learned model rather than the collected data. As a result, the learned model is only pessimistic and only in state-actions where there is no data. This treatment makes intuitive sense in the offline RL setting since the agent is not allowed to interact with the environment to explore and correct for optimistic mistakes.

Theoretically, Uehara & Sun (2021) showed that for sufficiently accurate learned dynamics model on the offline data distribution (e.g., as measued by $\mathbb{E}_{(s,a)\sim\mathcal{D}}D_{TV}[\hat{M}^{\text{MLE}}(\cdot|s,a)||\hat{M}(\cdot|s,a)] \leq \epsilon$ where $\hat{M}^{\text{MLE}}$ is the MLE dynamics), simultaneously minimizing model advantage and optimizing policy on the learned dynamics model yields a policy that is competitive with any policy found on the offline dataset. This can be done by setting the $\lambda$ parameter to be sufficiently high in the RAMBO algorithm.

Empirically, Rigter et al. (2022) showed that RAMBO achieves the best overall performance amongst other model-based and model-free baselines in four MuJoCo tasks with varying dataset qualities and it significantly

outperforms other model-based baselines in the AntMaze environment which has more challenging contact dynamics. Compared with COMBO (Yu et al., 2021), a SOTA model-based offline RL algorithm, RAMBO learns a smoother dynamics model, potentially making it less prone to local optima.

> **Robust Control Summary**: Robust control approaches reveal a hidden insight behind value-prediction approaches: in the process of finding value-equivalent MDPs, the dynamics model is actually becoming optimistic on some state-action pairs and pessimistic on others depending on its visitation. The inherent pessimism makes robust control approaches especially suited for addressing distribution-shift, such as in offline RL.

### 4.4 Differentiable Planning

Instead of explicitly defining model learning objectives, differentiable planning approaches embed the policy (or trajectory) optimization process with respect to the learned model as a differentiable program and update the model with respect to a higher level objective, such as maximizing return in the true environment or the standard Bellman error loss using environment samples (Mnih et al., 2013). These approaches typically take on a bi-level optimization format where optimality with respect to the learned model is defined as a constraint.

Farquhar et al. (2018) first recognized that for discrete actions and deterministic dynamics, the multi-step look-ahead search in VPN (Oh et al., 2017) can be interpreted as an neural network layer, which can be used to process state inputs before outputting the final value prediction. They proposed the TreeQN architecture which can be formulated as the following bi-level optimization problem:

$$\min_{\hat{M},Q} \mathbb{E}_{(s,a,s')\sim\mathcal{D}} \left(R(s,a) + \gamma V(s') - Q(s,a)\right)^2$$
$$\text{s.t. } V(s) = \arg\max_{a_{0:K}} \mathbb{E}_{\hat{M}} \left[ \sum_{k=0}^{K-1} \gamma^k R(s_k, a_k) + \gamma^K V(s_K)|s_0 = s \right] . \tag{TreeQN 41}$$

In a box-pushing environment, the authors showed that TreeQN significantly outperformed DQN (Mnih et al., 2013) and its performance improved when increasing the look-ahead depth from 1 to 3 steps. In the Atari environments, TreeQN outperformed DQN in all except 1 environment and consistently achieved higher performance with fewer environment steps. However, the effect of look-ahead depth in Atari were not observable.

Nikishin et al. (2022) proposed a similar bi-level optimization approach called Optimal Model Design (OMD). However, instead of using a multi-step look-ahead search, the constraint was defined using the first-order optimality condition for Bellman error minimization with respect to the learned model. The OMD model and policy objectives are defined as follows:

$$\min_{\hat{M},Q} \mathbb{E}_{(s,a,s')\sim\mathcal{D}} \left(R(s,a) + \gamma V(s') - Q(s,a)\right)^2$$
$$\text{s.t. } \nabla_{\hat{M}} \mathbb{E}_{(s,a)\sim\mathcal{D}, s'\sim\hat{M}(\cdot|s,a)} \left[ \left(R(s,a) + \gamma V(s') - Q(s,a)\right)^2 \right] = 0 , \tag{OMD 42}$$

where the soft-value function $V(s) := \log \sum_a \exp(Q(s,a))$ was used to make the constraint differentiable.

A novelty of the OMD approach is the use of the implicit function theorem to differentiate through the constraint. This method was later extended in (Bansal et al., 2023) where only a metric on the model loss was included in the upper objective to automatically weight task-relevant features while the standard prediction objective under the optimized metric was still used for model learning and formulated as an additional constraint. Sharma et al. (2023) used a similar implicit differentiation approach in a setting where the reward function is sampled from a distribution. Wang et al. (2021) used a sample-based approximation of implicit gradients in the setting of predicting missing parameters in an MDP from logged trajectories and then perform offline RL.

Empirically, Nikishin et al. (2022) showed that OMD outperformed MLE in Cartpole and one MuJoCo environment when the dynamics model hidden dimensions or parameter norm were constrained and when distracting state dimensions were added.

In contrast to the above value-based upper objectives, Amos & Yarats (2020) proposed to re-parameterize the policy by embedding the model in a differentiable Cross Entropy Method (DCEM) trajectory optimizer. Each action outputted by the DCEM policy is computed by running the DCEM trajectory optimization algorithm for a finite number of iterations. Then, the dynamics model parameters are updated with respect to the upper level objective of the expected return of the policy in the true environment. The DCEM model and policy objectives are defined as follows:

$$\max_{\hat{M}} \mathbb{E}_{(s,a) \sim d_M^\pi}[R(s,a)]$$
$$\text{s.t. } \pi(a|s;\hat{M}) = \arg\max_{a_{0:K}} \mathbb{E}_{\hat{M}} \left[ \sum_{k=0}^{K} R(s_k, a_k)|s_0 = s, a_0 = a \right]. \tag{DCEM 43}$$

The upper level objective is optimized using the Proximal Policy Optimization algorithm (Schulman et al., 2017b).

Empirically, the authors showed that DCEM matched the performance of Dreamer (Hafner et al., 2019), a popular MBRL algorithm, on two MuJoCo tasks with an order-of-magnitude fewer environment samples.

Prior to the above deep learning based differentiable planners, Joseph et al. (2013) and Bansal et al. (2017) explored similar bi-level optimization formulations of model learning where the model parameters are updated using finite-difference gradient approximation and Bayesian optimization, respectively. These methods were applied to simple and low-dimensional problems and they are difficult to scale to high-dimensional settings with large dynamics model parameterizations.

Recently, Gehring et al. (2021) theoretically studied differentiable planners in a simplified reward prediction setting and found that their learning dynamics can be understood as pre-conditioned gradient descent, which can significantly increase convergence speed depending on the initialization of the value estimate.

Despite limited applications in RL, differentiable planners are a popular approach for inverse RL and imitation learning (Tamar et al., 2016; Okada et al., 2017; Srinivas et al., 2018; Karkus et al., 2017; Amos et al., 2018). Aiming to improve the efficiency of learning differentiable planners, Bacon et al. (2019) proposed a Lagrangian method which bypasses solving the lower optimization problem for every upper optimization step. Leveraging the relationship between value-equivalence and robust control (i.e., (38)), Wei et al. (2023) showed that favorable properties of differentiable inverse planners can be attributed to learning pessimistic dynamics and robust policies.

> **Differentiable Planning Summary**: Differentiable planning represents the most direct approach to solving objective mismatch by embedding a minimal set of standard MBRL components (e.g., only a model and a policy) in a differentiable program so that all components optimize the same objective. This class of approaches bypasses return estimation and focuses directly on return optimization. It also involves the least amount of human intervention in the objective design process.

## 5 Discussion

This paper reviewed solutions to the objective mismatch problem in MBRL and classified the approaches into a taxonomy based on their structure and intuition. The taxonomy highlights that these approaches have influences on key components of MBRL which we discuss below:

- MBRL objective design and the search for value optimization-equivalence in both the model and the policy.

- Important agent properties, such as ground truth model identifiability, agent exploration behavior, and model optimism and pessimism.

Table 1: Comparisons of architecture and implementation of the core reviewed decision-aware MBRL algorithms. For each algorithm, we list its objective mismatch solution category (DC=distribution correction, MS=model-shift, PS=policy-shift, CAI=control-as-inference, VE=value-equivalence, VP=value-prediction, RC=robust control, DP=differentiable planning), dynamics model type, whether ensembling of dynamics was used, whether the dynamics model was trained to make multi-step predictions, the policy optimization algorithm, and other agent components.

| Algorithm | Category | Model type | Ensemble | Multi-step | Policy algo. | Other components |
|---|---|---|---|---|---|---|
| DAM (Haghgoo et al., 2021) | DC-MS | Mixture density network | No | No | Shooting | Discriminator |
| TOM (Ma et al., 2023) | DC-PS | Mixture density network | Yes | No | SAC | Discriminator, model $\tilde{Q}$ |
| AMPL (Yang et al., 2022) | DC-PS | Gaussian | Yes | No | TD3 | Discriminator, IW estimator |
| GAMPS (D'Oro et al., 2020) | DC-PS | Linear Gaussian | No | No | PG | - |
| VMBPO (Chow et al., 2020) | CAI | Gaussian | Yes | No | SAC | Discriminator |
| MNM (Eysenbach et al., 2022) | CAI | Gaussian | Yes | No | SAC | Discriminator |
| ALM (Ghugare et al., 2022) | CAI | Latent Gaussian | No | Yes | DDPG | Discriminator, encoder |
| VAML (Farahmand et al., 2017) | VE-VP | Tabular | No | No | VI | Adversarial V |
| MML (Voloshin et al., 2021) | VE-VP | Gaussian | No | No | Discrete SAC | Adversarial V & IW |
| MuZero (Schrittwieser et al., 2020) | VE-VP | Latent deterministic | No | Yes | DQN, MCTS | - |
| VPN (Oh et al., 2017) | VE-VP | Latent deterministic | No | Yes | DQN, TD search | - |
| TD-MPC Hansen et al. (2022; 2023) | VE-VP | Deterministic | No | Yes | MPPI | Encoder |
| VE (Grimm et al., 2020) | VE-VP | Deterministic | No | No | DQN | - |
| VaGradM (Voelcker et al., 2022) | VE-VP | Deterministic | Yes | No | SAC | - |
| PAML (Abachi, 2020) | VE-VP | Deterministic | No | Yes | DDPG | - |
| LAMPS (Vemula et al., 2023) | VE-RC | Gaussian | Yes | No | SAC | - |
| RAMBO (Rigter et al., 2022) | VE-RC | Gaussian | Yes | No | SAC | - |
| TreeQN (Farquhar et al., 2018) | DP | Deterministic | No | No | DQN | - |
| OMD (Nikishin et al., 2022) | DP | Deterministic | No | No | Discrete SAC | - |
| DCEM (Amos & Yarats, 2020) | DP | Latent deterministic | No | No | CEM | - |

Table 2: Comparisons of evaluation and experiments of the core reviewed decision-aware MBRL algorithms. For each algorithm, we list its main evaluation environments, main baseline algorithms, the types of model misspecification evaluated, the metric used for evaluating model exploitation (policy gradient or Q estimation accuracy), and whether the it's designed for the online or offline RL setting.

| Algorithm | Environments | Baselines | Misspecification | Exploitation | On/offline |
|---|---|---|---|---|---|
| DAM (Haghgoo et al., 2021) | Driving | MLE | Multimodal | - | Online |
| TOM (Ma et al., 2023) | MuJoCo | MBPO, PDML, VaGradM | - | - | Online |
| AMPL (Yang et al., 2022) | Maze2D, MuJoCo, Adroit | COMBO, CQL | - | - | Offline |
| GAMPS (D'Oro et al., 2020) | MuJoCo | PG | Remove features | PG | Offline |
| VMBPO (Chow et al., 2020) | MuJoCo | MBPO, SAC | - | - | Online |
| MnM (Eysenbach et al., 2022) | MuJoCo, DMC, Metaworld, ROBEL | MBPO, SAC | Low-res. | Q | Online |
| ALM (Ghugare et al., 2022) | MuJoCo | MBPO, SAC-SVG | - | Q | Online |
| VAML (Farahmand et al., 2017) | Tabular | MLE | Low-res. | Q | Online |
| MML (Voloshin et al., 2021) | LQR, Cartpole | MLE, VAML | - | Q | Offline |
| MuZero (Schrittwieser et al., 2020) | Atari, Go, Chess, Shogi | AlphaZero | - | - | Online |
| VPN (Oh et al., 2017) | Atari | DQN, MLE | - | - | Online |
| TD-MPC (Hansen et al., 2022; 2023) | DMC, Metaworld | SAC, Dreamer-v3 | - | - | Online |
| VE (Grimm et al., 2020) | Four rooms, Catch, Cartpole | MLE | Rank | - | Online |
| VaGradM (Voelcker et al., 2022) | MuJoCo | MBPO, VAML | NN size, distractor | - | Online |
| PAML (Abachi, 2020) | MuJoCo | MLE, DDPG | Distractor | - | Online |
| LAMPS (Vemula et al., 2023) | MuJoCo | MBPO | - | - | Online |
| RAMBO (Rigter et al., 2022) | MuJoCo, AntMaze | COMBO, CQL | - | Q | Offline |
| TreeQN (Farquhar et al., 2018) | Atari | DQN, A2C | - | - | Online |
| OMD (Nikishin et al., 2022) | MuJoCo, Cartpole | MLE | NN size | - | Online |
| DCEM (Amos & Yarats, 2020) | DMC | Dreamer | - | - | Online |

- Optimization approaches and how to extract maximum information from novel objectives.

- Downstream applications and re-using dynamics models.

- Benchmarking and improving rigor in MBRL.

### 5.1 Decision-Aware Objective Design

The proposed taxonomy suggests a single principle to decision-aware model and policy objective design: *value optimization-equivalence*, where both the model and the policy should be trained to optimize the expected return in the real environment. The value optimization-equivalence principle extends the value-equivalence principle, a previously proposed principle for MBRL (Grimm et al., 2020; 2021; 2022) and also reviewed in Section 4.3.1, in two ways. First, it suggests that in addition to decision-aware model learning, which was proposed in VE, policy learning should also be decision-aware, or rather model-aware. Second, model learning can focus on value optimization (e.g., control-as-inference and differentiable planning) rather than just value or Bellman operator estimation (e.g., distribution correction and value-equivalence). Below, we discuss how the value optimization-equivalence principle is manifested in the reviewed approaches. We identify a split among the reviewed approaches similar to the policy-based vs. value-based and model-free vs. model-based paradigms in RL and whether these approaches are designed to explicitly handle errors occurring at different stages of the learning process.

**Error-awareness:** Error-aware approaches in the distribution correction and control-as-inference categories achieve value optimization-equivalence by explicitly modeling the errors of simulated trajectories using distribution correction and adapting the model learning and policy optimization objective to better approximate policy performance in the real environment. These error-modeling schemes fit imperfect models to relevant data akin to locally weighted learning (Atkeson et al., 1997). Methods that only address model-shift, e.g., by redesigning policy objectives, can be more permissive towards the specific model-training method. However, authors have also identified model-training objectives with better optimization behavior from the policy evolution, policy-gradient matching, variance reduction, or control-as-inference perspectives.

On the other hand, error-agnostic approaches achieve value-optimization equivalence by constructing equivalent classes of MDP dynamics, either explicitly as in value-equivalence or implicitly as in differentiable planning, such that evaluating policies in the learned dynamics is close to that of the true dynamics. These approaches simplify the training pipeline as they do not alter the standard model-based value estimation procedure, for example by adding density-ratio estimators and auxiliary model-based reward bonus as in DAM and MNM.

However, many approaches in both the error-aware and error-agnostic categories still do not explicitly control for policy-shift (i.e., limiting policy divergence in (6)) or introduce policy-shift to new agent components. In error-aware approaches, the value optimization-equivalence property may be hindered if the density-ratio estimator trained on past data cannot generalize to new policies. Similarly, in error-agnostic approaches, minimizing the value-prediction loss on past data may not ensure small loss on new data. Instead, policy-shift is implicitly addressed by taking small optimization steps and relying on fast alternation between model and policy updates (Ma et al., 2023), optimizing against the worst-case density-ratio (Voloshin et al., 2021), against the worst-case dynamics (Rigter et al., 2022), or introducing reward bonus (Haghgoo et al., 2021; Eysenbach et al., 2022; Yang et al., 2022).

**Value-awareness:** In RL, policy-based and value-based approaches are distinguished by whether a value function is a necessary component of the algorithms, and policy-based approaches can operate on Monte Carlo return estimators rather than explicit value function estimators. Using this analogy, we can classify distribution correction and differentiable planning as policy-based and control-as-inference and value-equivalence as value-based. Similar to policy optimization in RL, value-based approaches for model learning are tied to the quality of the value function estimator, which can require special care either in objective design (e.g., identifying the extrapolation errors of value function estimates (Voelcker et al., 2022; Farahmand et al., 2017; Voloshin et al., 2021)) or implementation (e.g., making sure the value function estimator is sufficiently accurate before applying to model learning (Eysenbach et al., 2022)).

**Model-awareness:** The value optimization-equivalence principle also implies a hybridization of model-free and model-based RL approaches, which is manifested in most reviewed works. The traditional view of model-based approaches is to build an as accurate as possible model of the environment. The value optimization-equivalence principle suggests to instead stay as close as possible to model-free RL in the sense that sampling from the model or evaluating samples from the model is close to having samples drawn from the true environment. This is intuitive because model-free RL evaluates returns on true samples; this is as if we were running MBRL with privileged knowledge of the true dynamics. However, there are still several advantages of MBRL that are beyond just having better environment sample complexity. As Gehring et al. (2021) suggested, value optimization-equivalent MBRL likely enjoys optimization advantages due to the relationship with accelerated gradient descent methods. Value optimization-equivalent MBRL may also enjoy exploration and safety advantages depending on whether and in what part of the state-action space the learned dynamics is optimistic or pessimistic.

## 5.2 Properties of Decision-Aware Agents

Decision-aware MBRL methods can mirror the enactive view of cognitive science (Di Paolo & Thompson, 2014), where the role of perception is not to build a true model of the environment but to serve as an interface between the agent and the environment to enable flexible and adaptive behavior. Under this view, the learned model is free to deviate from the environment. It is thus useful to understand how learned models deviate from the true model, how agents behave during the learning process, and implications for downstream applications.

The fullest picture is depicted by Vemula et al. (2023) who showed in an exact decomposition of the return gap that the learned model is optimistic in the in-distribution region of the state-action space and pessimistic in the out-of-distribution region. The adversarial model learning approach by Rigter et al. (2022) adopts the pessimistic perspective, but instead learns an objective (accurate) model in the in-distribution region. Agents trained with this objective will likely be more conservative in their exploration behavior and less prone to model-exploitation. In contrast, control-as-inference approaches learn an optimistic model of the dynamics in the in-distribution region, and they do not explicitly model out-of-distribution behavior in the model learning objective. Instead, control-as-inference learns conservative behavior in the policy-optimization process by encouraging agents to follow well-predicted states and only deviate when additional information about the dynamics model can be gained. These agents will likely exhibit more exploratory behavior than robust control agents in a more structured manner. For example, Eysenbach et al. (2022) found that their agent tends to make the goal positions appear closer than they actually are in model rollouts so that the goals are easier to reach, which is beneficial in sparse reward settings to accrue more feedback for the policy. Control-as-inference agents also represent an interesting point of contact with recent information-theoretic approaches to agent objective design in both task-driven and unsupervised RL (Hafner et al., 2020; Rhinehart et al., 2021).

A substantial question for value optimization-equivalent agents is whether the true environment model can be identified. The findings of Nikishin et al. (2022) suggest that the true environment model is not identifiable given that two different dynamics models can lead to the same optimal policy. This equivalence is related to the research on state-abstraction in MDPs where functionally similar states are aggregated (in the current case ignored) in order to avoid modeling task-irrelevant state features (Li et al., 2006). One way to improve identifiability is to add an environment or reward prediction loss as in (Schrittwieser et al., 2020; 2021), however, this introduces a trade-off between value-equivalent vs. accurate models which has to be specified by the modeler. Nikishin et al. (2022) suggested that unidentifiability is likely a blessing rather than a curse because there are likely value-equivalent models that are easier to learn than the true model. However, Wei et al. (2023) showed in an inverse RL setting with differentiable planners that favorable robustness properties of value-equivalent models may succumb to overly high inaccuracy. They further suggested regularizing the model's state-prediction accuracy. As noted by several authors, the set of value-equivalent models reduces with increasing number of policies and values considered (Grimm et al., 2020; Voloshin et al., 2021). This suggests that identifiability is possible in a multi-task or meta-learning setting and an experimental validation is provided in (Berke et al., 2022) from a cognitive science perspective. However, currently most works on decision-aware MBRL has focused on the single-task setting.

## 5.3 Optimization Approaches

In order to gain the full benefit from novel objective formulations designed to solve objective mismatch, different optimization techniques may be needed. We highlight two optimization approaches from the survey. The majority of reviewed works use manually designed objectives for model learning and policy optimization mostly by bounding the true return and leveraging existing optimization techniques such as policy-gradient and adversarial training. However, certain properties of decision-aware agents can be lost in manual formulation and optimization. For example, optimistic or pessimistic behavior characteristic of decision-aware agents is lost in models learned using value-prediction. In contrast to manual design of component objectives, differentiable planning approaches use a single objective of the true return and offload the complexity of optimization to differentiable programming. These approaches mostly take on a bilevel-optimization format, where an update of both the model and the policy need to take into account the optimal policy with respect to the current model. To this end, earlier differentiable planners such as (Farquhar et al., 2018; Amos & Yarats, 2020) relied on finite-horizon or truncated-horizon policy objectives where the gradients can be computed using backpropagation. More recent approaches such as (Nikishin et al., 2022; Bansal et al., 2023) have explored alternative optimization techniques such as implicit differentiation which can more precisely compute the gradient of infinite-horizon policies. However, the authors have also commented on the added complexity and approximation to these approaches. While manually designed optimization is advantageous for understanding the properties of decision-aware agents, more efficient end-to-end optimization approaches may be desirable for scaling and broadening application domains.

## 5.4 Downstream Applications

An important use case for MBRL is transfer learning, where the dynamics model learned in a source task can be used as an environment simulator for a target task to reduce or eliminate the number of environment samples (Taylor & Stone, 2009; Zhu et al., 2023). For traditional environment prediction-based model learning approaches, transferring would simply require the model to generalize in state-action space covered by the target task. However, transfer learning potentially presents a larger challenge for value optimization-equivalent agents because of model unidentifiability and the fact that decision-aware models are usually biased. In this setting, it is not clear whether model bias, especially the optimistic bias, which aids optimization in the source task is also help the target task. However, given that model-identifiability can be improved with multi-task training, decision-aware agents may be especially suited for tasks that require continuous model fine-tuning or adaptation such as in some instances of meta RL (Nagabandi et al., 2018b;a) and lifelong learning (Khetarpal et al., 2022). Within a limited set of multi-task decision-aware MBRL works, Tamar et al. (2016) and Karkus et al. (2017) showed that differentiable planners trained on a variety of imitation tasks led to promising task transfer capabilities. However, Karkus et al. (2017) observed that multi-task training did not contribute to learning more accurate models and the models were still "wrong yet useful". On the other hand, Hansen et al. (2023) achieved an almost two-fold improvement after fine-tuning the TD-MPC model pretrained on a significant 80 RL tasks but did not examine the accuracy of the learned model. Thus, the relationship between model accuracy, task transfer, and decision-aware MBRL is still an open question.

Another important utility of MBRL is to enhance the transparency and explainability of RL agents through the separation of model and policy, where designers can introspect the learned model to understand agent behavior and potentially correct agent failures. An important aspect of transparency is that the agent designer can easily comprehend and identify sub-optimalities in the model and make precise editing decisions (Räuker et al., 2023). The biases and unidentifiability in decision-aware agents introduce significant challenges for model comprehension since the true environment is no longer the ground truth or the optimization target and it may not be appropriate to correct the model towards the ground truth only on some identified states but not others since it may alter the learned value-equivalence.

These application considerations suggest a need to better understand the properties of decision-aware MBRL agents, such as value-equivalent MDPs, in order to reap their benefits without losing that of traditional MBRL.

## 5.5 Evaluations and Benchmarks

While resolving objective mismatch focuses on designing novel agent objectives or training procedures, the qualities of the objectives should be measured based on agent behavior. Since the ultimate goal of decision-aware agents is to achieve high returns, the final performance and learning speed (i.e., the number of environment steps to reach a performance threshold) are the primary evaluation metrics which have been used by most reviewed decision-aware MBRL works. However, more fine-grained evaluation of decision-aware MBRL should probe its advantages over traditional MBRL and model-free RL, most importantly, robustness to model misspecification and model-exploitation in both online and offline RL, which are the top two motivators for most reviewed methods backed by theory (6). Many reviewed works such as (D'Oro et al., 2020; Eysenbach et al., 2022; Farahmand et al., 2017; Voelcker et al., 2022; Abachi, 2020; Nikishin et al., 2022; Grimm et al., 2020) compared agent performance under different levels of model misspecification by varying the capacity of the dynamics models, removing state features, or adding distracting state features (see Table 2). Evaluation protocols of model-exploitation have also been developed in recent years by measuring value-estimation bias (Chen et al., 2021; Fujimoto et al., 2018). We recommend future work on decision-aware MBRL to include these experiments.

Beyond model misspecification and exploitation, some authors also probed more specific agent properties (e.g., exploration behavior). For example, Eysenbach et al. (2022) and Rigter et al. (2022) assessed the optimism and smoothness, respectively, of their learned dynamics models through visualizations in selected environments. Since these properties do not necessarily apply to all decision-aware MBRL approaches, different works performed different evaluations in different environments, which makes comparison of approaches challenging if not incomplete. Thus, as our understanding of the properties and utilities of decision-aware agents continues to develop and mature, we may benefit from documenting the agent properties that each environment is designed to test and developing behavior suites (Osband et al., 2019) for decision-aware agents.

The consistency between theory and implementation in decision-aware MBRL also warrants additional attention. Several works have remarked on the theory-implementation gap (Eysenbach et al., 2022; Modhe et al., 2021; Lovatto et al., 2020), which are mostly due to additional moving components and optimization challenges (e.g., requirements on value estimation accuracy as a result of including value estimates in model training objectives). Thus, evaluations of decision-aware MBRL should focus on not only the final performance but also implementation easiness, training stability, debugging tools (Lambert, 2021), sharing trained models (Pineda et al., 2021), and other optimization failure modes.

## 6 Conclusion

While resolving objective mismatch promises to boost the capability of MBRL agents, how to design aligned objectives for different agent components remains an open question. To this end, we found that all current efforts to address the objective mismatch problem can be understood along the lines of 4 major categories: *distribution correction, control-as-inference, value-equivalence, and differentiable planning*. These efforts point to a single principle for designing decision-aware objectives: *value optimization-equivalence*. Under this principle, both the dynamics model and the policy should be trained to optimize the expected return, effectively achieving a hybridization of model-free and model-based RL. We recommend future work to continue to enhance our understanding of decision-aware agents, identify their practical utilities, and design appropriate evaluation suites to fully harvest their benefits.

## Broader Impact

This paper synthesizes theoretical and empirical results for solving objective mismatch towards building more capable MBRL agents. RL and automated decision-making system can have important ramifications, especially when directly interfacing with humans. A major subset of these ramifications stems from misspecified reward and training environments. While we have focused on correctly specified reward and training environments, we remark that identifying and debugging these misspecifications in decision-aware MBRL

agents is likely harder than traditional RL agents. We believe developing better theoretical understanding and empirical testing suites are a first step towards the transparency required to mitigate harms from decision-aware MBRL agents.

## Acknowledgments

This work was partly supported by the German Research Foundation (DFG, Deutsche Forschungsgemeinschaft) as part of Germany's Excellence Strategy – EXC 2050/1 – Project ID 390696704 – Cluster of Excellence "Centre for Tactile Internet with Human-in-the-Loop" (CeTI) of Technische Universität Dresden, and by Bundesministerium für Bildung und Forschung (BMBF) and German Academic Exchange Service (DAAD) in project 57616814 (SECAI, School of Embedded and Composite AI).

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
