# OpenReview forum: "A Unified View on Solving Objective Mismatch in Model-Based Reinforcement Learning"
_TMLR — Accepted by TMLR_

### Review · Reviewer_ic6a · 2023-11-29

**Summary Of Contributions:**

The paper provides a survey and taxonomy of model-based RL methods that learn a model of the system dynamics in a task-oriented manner. Four categories of approaches are identified, with their respective characteristics and objective functions. In total, 43 papers are surveyed. The proposed taxonomy suggests *value optimization-equivalence* as an important design principle, by which "the model and the policy should be trained to optimize the expected return in the real environment".

**Audience:**

Yes

**Broader Impact Concerns:**

The paper poses no ethical concerns

**Claims And Evidence:**

Yes

**Requested Changes:**

Please see my detailed comments above. Below I just repeat my requests in a short form.

1) Scope and Related Work

    a) position the paper w.r.t. the work on adaptive optimal control with regards to the objective mismatch

    b) [critical] either add active inference as a category (or sub-category of control-as-inference (e.g., [1])) or explain why it is out-of-scope in the Related Work section

    c) mention meta-learning and explain why it is not included


2) [critical] Objective Mismatch: consider reframing the paper, mentioning it as one possible problem instead of the "root cause". Or alternatively provide a theorem or strong empirical evidence that it is indeed the "root cause".

3) Decision-Aware MBRL vs Multi-Task MBRL: in the introduction, discuss the motivation behind focusing on decision-aware MBRL for a single task, as the considered methods do.

4) [critical] Value Optimization-Equivalence: show how this principle is derived from the taxonomy and how each category of methods obeys/implements this principle.

### References
[1] Imohiosen, A., Watson, J., & Peters, J. (2020). Active inference or control as inference? A unifying view. In Active Inference: First International Workshop, IWAI 2020, Co-located with ECML/PKDD 2020, Ghent, Belgium, September 14, 2020, Proceedings 1 (pp. 12-19). Springer International Publishing.

**Strengths And Weaknesses:**

## Strengths
- The paper strikes a good balance between the depth and breadth of the review
- The identified clusters of approaches provide a valuable vocabulary and a map for navigating the field
- The insights given at the end of each subsection and in the Discussion section provide a number of sharp and clear distinctions and concepts for delineating different approaches

## Weaknesses
### 1. Scope and Related Work
The chosen scope is somewhat restrictive and might need further clarification. The paper explicitly states that it collected the papers for review "by examining articles citing (Lambert et al., 2020)". However, the problem of learning a dynamics model and optimizing a policy has been studied quite extensively within *adaptive optimal control* or *dual control*. Perhaps stating in the Related Work section how the considered methods fit into the bigger picture would be helpful.

The paper understandably had to limit the amount of related work considered. However, I feel it should at least mention the omitted prominent approaches. For example, *active inference* is notorious for arguing for learning the model and the controller jointly, both of which are minimizing a single objective function — the variational free energy (see e.g. [1]). Furthermore, meta-learning or Bayes-adaptive MDPs are also approaching the same problem, e.g., [2].


### 2. Framing
Currently, the paper relies on "objective mismatch" introduced by (Lambert et al., 2020) as the starting point — an observation that the model learning objective (e.g., maximum likelihood) is different from the policy learning objective (maximizing return). The referenced paper argues that the "objective mismatch" is the **cause** of the phenomenon that a better accuracy in one-step model prediction does not necessarily lead to better policies. However,
1) it does not seem to be a non-controversial fact that it is the "objective mismatch" that is the cause of the observed phenomenon (e.g., see the discussion of the referenced paper on OpenReview https://openreview.net/forum?id=Sked_0EYwB (note that the accepted L4DC version http://proceedings.mlr.press/v120/lambert20a/lambert20a.pdf is very similar to the rejected ICLR https://openreview.net/pdf?id=Sked_0EYwB)).
2) Indeed, in [3], it is shown that even a perfect model does not necessarily improve sample efficiency of RL.
3) As the authors of the present paper note in Sec. 2.3, the issue may be due to compounding errors from using learned one-step prediction models.
4) Additionally, the overall performance depends on what type of RL or planning method is used on top of the learned model.

Therefore, it appears too strong a statement to say that "objective mismatch" is the **main reason** why a learned model is not helping the policy learning so much. But luckily, I think it is also not necessary for this paper to argue this way. It is sufficient to note that such behavior has been observed and that it can be attributed to multiple factors, one of which is the objective mismatch.


### 3. Decision-Aware MBRL vs Multi-Task MBRL
The relation between decision-aware MBRL and multi-task learning is discussed in one paragraph in Sec. 5.4, but I think this could be discussed in a bit more detail and also mentioned in the introduction. Namely, a key advantage of learning a model is that it can be reused for other tasks. However, if one would learn a task-specific dynamics model, as the surveyed paper are trying to do, then what is the benefit of that model — i.e., why not directly learn model-free on that task? I think it would be worth providing some arguments addressing this question somewhere in the beginning of the paper.


### 4. Value Optimization-Equivalence
1) In Sec. 5.1, the paper immediately claims that "The proposed taxonomy suggests a single principle to decision-aware model and policy objective design: *value optimization-equivalence*". However, it is not shown how this is derived or where it follows from. For example, it is not clear how methods of "Distribution Correction" category (Sec. 4.1) and "Control-As-Inference" category (Sec. 4.2) are implementing the *value optimization-equivalence* principle.

2) Furthermore, if it is indeed the underlying principle, then it is strange that the category of "Value-Equivalence" methods (Sec. 4.3) — which is explicitly implementing this principle — is placed at the same level as all other categories. Shouldn't it be at the root of the hierarchy in this case?


### References
[1] Tschantz, A., Millidge, B., Seth, A. K., & Buckley, C. L. (2020). Reinforcement learning through active inference. arXiv preprint arXiv:2002.12636.

[2] Zintgraf, L., Shiarlis, K., Igl, M., Schulze, S., Gal, Y., Hofmann, K., & Whiteson, S. (2019, September). VariBAD: A Very Good Method for Bayes-Adaptive Deep RL via Meta-Learning. In International Conference on Learning Representations.

[2] Palenicek, D., Lutter, M., Carvalho, J., & Peters, J. (2022, September). Diminishing Return of Value Expansion Methods in Model-Based Reinforcement Learning. In The Eleventh International Conference on Learning Representations.

---

> ### Author Response · Authors · 2023-12-21
>
> We thank the reviewer for the detailed review of the paper and the valuable feedback. We have made corresponding changes in the paper highlighted in blue. Below, we address the reviewer's comments in a point-by-point manner.
>
> **Weakness 1**: The chosen scope is somewhat restrictive and might need further clarification. The paper explicitly states that it collected the papers for review "by examining articles citing (Lambert et al., 2020)''. However, the problem of learning a dynamics model and optimizing a policy has been studied quite extensively within adaptive optimal control or dual control. Perhaps stating in the Related Work section how the considered methods fit into the bigger picture would be helpful.
>
> **Our response**: We thank the reviewer for highlighting these related research areas. Dual control, and the related topics of hidden parameter MDP and meta RL, are especially relevant to the objective mismatch problem and their typical solutions can be interpreted as error aware in the sense that the policy optimizer is aware of the uncertainty/error in the model. However, as the standard formulation of meta model learning still optimizes environment prediction error, we believe the objective mismatch issue is still present and prior work considering objective mismatch in these domains is even more limited.
> We've clarified our inclusion criteria in section 3.
>
> **Weakness 2**: The paper understandably had to limit the amount of related work considered. However, I feel it should at least mention the omitted prominent approaches. For example, active inference is notorious for arguing for learning the model and the controller jointly, both of which are minimizing a single objective function — the variational free energy (see e.g. [1]). Furthermore, meta-learning or Bayes-adaptive MDPs are also approaching the same problem, e.g., [2].
>
> **Our response**: We agree with the reviewer that active inference and meta RL are important related work to objective mismatch. We have added discussions of both of these classes of approaches to the related work section. For meta RL, we exclude it from the taxonomy because most meta RL algorithms are model-free and a handful of model-based ones are optimized for observation reconstruction (e.g. VariBAD). For active inference, we do not list it as its own category, since the derivation of the expected free energy objective function for action planning in active inference is still unsettled [1]. Nevertheless, we highlighted a connection between the ALM objective discussed in section 4.2 and the expected free energy objective, which was made implicit in the previous draft.
>
> [1] Millidge, B., Tschantz, A., & Buckley, C. L. (2021). Whence the expected free energy?. Neural Computation, 33(2), 447-482.

---

> ### Author Response · Authors · 2023-12-21
>
> **Weakness 2**: Currently, the paper relies on "objective mismatch'' introduced by (Lambert et al., 2020) as the starting point — an observation that the model learning objective (e.g., maximum likelihood) is different from the policy learning objective (maximizing return). The referenced paper argues that the "objective mismatch" is the cause of the phenomenon that a better accuracy in one-step model prediction does not necessarily lead to better policies. However, it does not seem to be a non-controversial fact that it is the "objective mismatch" that is the cause of the observed phenomenon (e.g., see the discussion of the referenced paper on OpenReview https://openreview.net/forum?id=Sked_0EYwB (note that the accepted L4DC version http://proceedings.mlr.press/v120/lambert20a/lambert20a.pdf is very similar to the rejected ICLR https://openreview.net/pdf?id=Sked_0EYwB)).
> Indeed, in [3], it is shown that even a perfect model does not necessarily improve sample efficiency of RL.
> As the authors of the present paper note in Sec. 2.3, the issue may be due to compounding errors from using learned one-step prediction models.
> Additionally, the overall performance depends on what type of RL or planning method is used on top of the learned model.
> Therefore, it appears too strong a statement to say that "objective mismatch" is the main reason why a learned model is not helping the policy learning so much. But luckily, I think it is also not necessary for this paper to argue this way. It is sufficient to note that such behavior has been observed and that it can be attributed to multiple factors, one of which is the objective mismatch.
>
> **Our response**: Thank you for suggesting the paper on the diminishing return of value expansion methods. We agree that objective mismatch is not the only reason why a learned model is not helping policy learning. We think that paper adequately highlights the complex interaction between model and policy quality when the two are measured by different objective functions. That being said, we think objective mismatch and the related model exploitation are fundamental issues most clearly highlighted by the offline RL setting.
>
> To properly acknowledge these nuance, we have revised the text in section 2.3 to state that "the complex interaction between model learning and policy optimization highlights the gap in the fundamental understanding of the misalignment between currently separate model and policy objectives.''
>
> **Weakness 3**: The relation between decision-aware MBRL and multi-task learning is discussed in one paragraph in Sec. 5.4, but I think this could be discussed in a bit more detail and also mentioned in the introduction. Namely, a key advantage of learning a model is that it can be reused for other tasks. However, if one would learn a task-specific dynamics model, as the surveyed paper are trying to do, then what is the benefit of that model — i.e., why not directly learn model-free on that task? I think it would be worth providing some arguments addressing this question somewhere in the beginning of the paper.
>
> **Our response**: Upon suggestions from other reviewers, we realized that we omitted a few related work on multi-task and transfer learning in decision-aware MBRL in the previous draft. We have added these to section 5.4. We agree that model transfer is an underexplored topic in MBRL. Our intention in this paragraph was to highlight the additional challenge brought to model transfer by decision-aware MBRL and the open questions.
>
> **Weakness 4**: In Sec. 5.1, the paper immediately claims that "The proposed taxonomy suggests a single principle to decision-aware model and policy objective design: value optimization-equivalence". However, it is not shown how this is derived or where it follows from. For example, it is not clear how methods of "Distribution Correction" category (Sec. 4.1) and "Control-As-Inference" category (Sec. 4.2) are implementing the value optimization-equivalence principle.
>
> Furthermore, if it is indeed the underlying principle, then it is strange that the category of "Value-Equivalence" methods (Sec. 4.3) — which is explicitly implementing this principle — is placed at the same level as all other categories. Shouldn't it be at the root of the hierarchy in this case?
>
> **Our response**: We have revised section 5.1 to add more context to the observed value optimization-equivalence principle. First, value optimization-equivalence extends value-equivalence by including objective design principle for both model learning and policy optimization and for return optimization rather than just constructing equivalent Bellman operators. Then in the error-awareness section, we discuss how the value optimization-equivalence principle is manifested in the four different solution categories.

---

> ### Author Response · Authors · 2023-12-21
>
> **Requested changes**:
> 1. Scope and Related Work
>
>     a) position the paper w.r.t. the work on adaptive optimal control with regards to the objective mismatch
>
>     b) [critical] either add active inference as a category (or sub-category of control-as-inference (e.g., [1])) or explain why it is out-of-scope in the Related Work section
>
>     c) mention meta-learning and explain why it is not included
>
> 2. [critical] Objective Mismatch: consider reframing the paper, mentioning it as one possible problem instead of the "root cause". Or alternatively provide a theorem or strong empirical evidence that it is indeed the "root cause".
>
> 3. Decision-Aware MBRL vs Multi-Task MBRL: in the introduction, discuss the motivation behind focusing on decision-aware MBRL for a single task, as the considered methods do.
>
> 4. [critical] Value Optimization-Equivalence: show how this principle is derived from the taxonomy and how each category of methods obeys/implements this principle.
>
> **Our response**: We believe we have addressed all requested changes in the revised paper and the above responses.

---

> > ### Comment · Reviewer_ic6a · 2023-12-25
> > **A few important remaining concerns**
> >
> > I thank the authors for their replies. The updates to the paper addressed most of my concerns. However, several points still need to be addressed.
> >
> > ## 1a) position the paper w.r.t. the work on adaptive optimal control with regards to the objective mismatch
> >
> > As a response, the authors added one paragraph at the end of Sec. 3 about Bayesian RL. However, I believe the authors' framing is not complete and it does not sufficiently place the work in the context. Therefore, clarifications are needed there and in other places in the paper.
> >
> > Namely, Bayesian RL in principle provides the complete solution to the "objective mismatch" problem: given a powerful parameterized model $p_\theta(s' | s, a)$, one would find a posterior $p(\theta | ...)$ that would only care to model those areas of the state-action space which are relevant for maximizing the reward (see Sec. 4 in the [Bayesian RL Survey](https://arxiv.org/abs/1609.04436) or https://youtu.be/5rev-zVx1Ps?feature=shared&t=3525). Purely from the information-theoretic perspective, Bayesian RL is the best one could ever hope to do in terms of efficiently learning the model and optimizing the policy jointly. Therefore, all the other approaches which split model learning from policy optimization or even don't learn a probabilistic model at all can be viewed as approximations of the Bayes-adaptive solution.
> >
> > I believe this point should be made clear in the paper, because it underlies the considered "objective mismatch" issue. E.g., the following statement should be reformulated
> >
> > > While it is **unclear** to what extent belief-space model learning and policy optimization alleviate the objective mismatch problem, there is also no study in this regard.
> >
> > because it is actually clear that Bayesian RL in principle provides the solution, but different implementations may introduce approximations that lead to the "objective mismatch" problem.
> >
> > ---
> >
> > Furthermore, the following statement should be adjusted:
> >
> > > We focused on (Lambert et al., 2020) because **it was the first work to identify** and name **the objective mismatch problem**
> >
> > Given the discussion above, it would be incorrect to claim that Lambert et al. (2020) was the first to "identify" this issue, because it is a well-known fact in dual control and Bayesian RL, and in RL in general it is known that learning task-specific state-abstractions is advantageous to learning full system dynamics, e.g., [1].
> >
> > Therefore, I would suggest to restrict the claim to saying, e.g., "Lambert et al., (2020) coined the term 'objective mismatch' in the context of MBRL to refer to the fact that in the common practice, the objective for model learning (i.e., maximum likelihood) is different from the objective for policy optimization (i.e., reward maximization) — hence the 'objective mismatch'. Among other factors, the objective mismatch was found to play an important role in explaining the empirical observation that the policy performance is only loosely correlated with the model quality. From the Bayesian RL perspective, this can be understood as..."
> >
> > [1] Li, L., Walsh, T. J., & Littman, M. L. (2006). Towards a unified theory of state abstraction for MDPs. AI&M, 1(2), 3.

---

> > > ### Author Response · Authors · 2023-12-29
> > >
> > > **Comment 1**: position the paper w.r.t. the work on adaptive optimal control with regards to the objective mismatch
> > >
> > > **Our response**: Thank you for highlighting this. We agree that Bayesian RL in principle solves the objective mismatch problem. To this end, we have added discussions about Bayesian model learning providing sufficient statistics to both model learning and policy optimization in section 2.3. In section 3, we have removed the statement on the extent to which belief-space planning and model learning alleviate the objective mismatch problem. We still exclude in-depth discussions of Bayesian RL from the survey since the majority of Bayesian RL focus on learning optimal exploration strategies as opposed to developing novel objectives for model and policy learning.
> > >
> > > **Comment 2**: Furthermore, the following statement should be adjusted: "We focused on (Lambert et al., 2020) because it was the first work to identify and name the objective mismatch problem."
> > >
> > > **Our response**: We agree that the word "identify" is not adequate in this context. We have revised the text to say that Lambert et al. (2020) coined the term "objective mismatch". We further state in section 2.3 that the term was coined because "in the common practice, the model learning objective (i.e., maximizing likelihood) is different from the policy optimization objective (i.e., maximizing return)".

---

> > > > ### Comment · Reviewer_ic6a · 2024-01-02
> > > > **All questions answered**
> > > >
> > > > I thank the authors for their response. All my questions got answered. I have no further reservations.

---

### Review · Reviewer_aW56 · 2023-12-04

**Summary Of Contributions:**

This paper provides a literature survey on the topic of objective mismatch in model-based reinforcement learning. Objective mismatch is the phenomenon that the dynamics model in MBRL is trained for a different objective and not aware of its role in the policy decision-making and learning; consequently, more accurate dynamics models often hurt policy performance. This paper's main contribution is a taxonomy of four categories of decision-aware MBRL approaches: Distribution Correction, Control-As-Inference, Value-Equivalence, and Differentiable Planning, which derive modifications to the model learning, policy optimization, or both processes for the purpose of aligning model and policy objectives and gaining better performance. Comparisons among different categories are presented and contextualized. Overall, this is a high-quality and sound survey of the literature on objective mismatch in MBRL.

**Audience:**

Yes

**Broader Impact Concerns:**

N/A.

**Claims And Evidence:**

Yes

**Requested Changes:**

1. Add some historical perspectives on objective mismatch in model learning, such as discussions on Locally Weighted Learning algorithms and their connections to the recent literature. While I believe this survey is high-quality and worthy of acceptance, I also do believe that proper historical contextualization and citations are important for a survey paper.

**Strengths And Weaknesses:**

Strengths:
1. This paper is very comprehensive in recent literature on decision-aware model-based reinforcement learning.

2. This paper is well-written and free of grammatical errors. The presentation for the main surveyed algorithms is very clear and to the sufficient level of details.

3. The summary and comparison for various different algorithmic categories are well motivated.

Weaknesses:
1. This paper primarily surveys only very recent papers, with most cited algorithms dating later than 2020. I believe there are several lines of classical literature on the subject that are worth mentioning. For example, there is a line of methods known as "locally weighted learning" or "locally weighted regression" in the 90s that have very similar flavor for model learning as the "distribution correction" approaches this survey studies.

Locally weighted learning: https://www.ri.cmu.edu/pub_files/pub1/atkeson_c_g_1997_1/atkeson_c_g_1997_1.pdf

---

> ### Author Response · Authors · 2023-12-21
>
> We thank the reviewer for the detailed review of the paper and the valuable feedback. We have made corresponding changes in the paper highlighted in blue. Below, we address the reviewer's comments in a point-by-point manner.
>
> **Weakness \& requested change**: This paper primarily surveys only very recent papers, with most cited algorithms dating later than 2020. I believe there are several lines of classical literature on the subject that are worth mentioning. For example, there is a line of methods known as "locally weighted learning" or "locally weighted regression" in the 90s that have very similar flavor for model learning as the "distribution correction" approaches this survey studies.
>
> Locally weighted learning: \url{https://www.ri.cmu.edu/pub_files/pub1/atkeson_c_g_1997_1/atkeson_c_g_1997_1.pdf}
>
> Add some historical perspectives on objective mismatch in model learning, such as discussions on Locally Weighted Learning algorithms and their connections to the recent literature. While I believe this survey is high-quality and worthy of acceptance, I also do believe that proper historical contextualization and citations are important for a survey paper.
>
> **Our response**: Thank you for highlighting this line of research. We have added discussions about locally weighted learning to section 5.1 in the discussion of error-awareness of distribution correction methods. We interpret locally weighted learning as fitting imperfect models only to relevant data.

---

### Review · Reviewer_e1y7 · 2023-12-10

**Summary Of Contributions:**

The authors present a summary of recent work related to objective mismatch in model-based reinforcement learning (MBRL). The collection of work is broken down into 4 categories, distribution correction, control-as-inference, value-equivalence, and differentiable planning. Contributions of each paper are summarized and compared to each other. The paper concludes with a significant discussion about various components of these methods for objective mismatch in MBRL.

**Audience:**

Yes

**Broader Impact Concerns:**

Broader impact seems sufficiently discussed.

**Claims And Evidence:**

Yes

**Requested Changes:**

# Necessary corrections

Eq. (33) to (37) are missing "min" on the rhs of the equation.

p.16, "Empirically, the authors showed that OMD [...]", this is ambiguous. Which authors? I presume this refers to Nikishin et al. but other work was cited since they were referenced last.

# Recommendations

Overall, I recommend making the distinction between optimal state/state-action value functions, state/state-action value functions and approximate state/state-action value functions explicit with separate notation, e.g., $Q^*, Q^\pi, \hat{Q}^*,  \hat{Q}^\pi$.

The paper might benefit from a summarization of the motivations of the compared methods, possibly in a similar style as tables 1 and 2 or, alternatively, as a few paragraphs in the discussion section. For instance, some papers seem to have a focus on the offline RL setting and some more on model-misspecification. Summarizing this information would help guide readers to the papers most relevant to the motivation behind their current research. Though it's also possible that there isn't enough differences between the stated (or inferred) motivations of each paper to be worthwhile, but it would be a nice addition if there is.

Eq (13), In what sense is $\tilde{Q}$ a dual? Maybe add a sentence or two to clarify.

Sec 4.2, in what way is $\mathcal{O}$ representing "optimality"? Is there any justification or explanation that could be summarized and included in this survey?

bottom p. 11, "While theoretical understanding of self-predictive objectives is still lacking [...]", Subramanian, et al. [1] provide some theoretical discussion about self-predictive objectives and provide conditions under these latent variables (which they call information states) satisfy a form of value-equivalence.

Control-as-inference summary, It's not clear to me what exactly is meant by hand-designed.

Sec 4.3.2 felt generally too sparse on details and discussion. For instance, in the robust control summary, It's wasn't clear to me where the discussion about pessimism came from.

p. 16, the name OMD was taken from Bacon, et al. [2]. In addition, it would be good to discuss this work as they take a different approach to implicit differentiation and instead consider using the Lagrangian to solve the Bi-level optimization problem.

p. 20, "we can classify distribution correction and differentiable planning as policy-based [...]", I'm not sure I would classify differentiable planning as being only policy-based or value-based. Some differentiable planning methods (e.g., OMD) learn a state-action value function.

p. 21, "decision-aware MBRL has focused on the single-task setting" & "transfer learning presents a larger challenge for value optimization-equivalent agents", I would argue the focus of value iteration networks (Tamar et al., 2016) and QMDP-Net (Karkus et al., 2017) is multi-task/transfer learning. The motivation for these approach is to generalize over task specification (e.g., a floor map).

# References

[1] Subramanian, J., Sinha, A., Seraj, R., & Mahajan, A. (2022). Approximate information state for approximate planning and reinforcement learning in partially observed systems. The Journal of Machine Learning Research, 23(1), 483-565.

[2] Bacon, P. L., Schäfer, F., Gehring, C., Anandkumar, A., & Brunskill, E. (2019). A Lagrangian method for inverse problems in reinforcement learning. In Optimization in RL workshop at NeurIPS (Vol. 2019).

**Strengths And Weaknesses:**

The paper does a good job contextualizing and summarizing various approaches for objective mismatch. This work puts forth a reasonable categorization of these methods and should provide a good entry point for researcher wanting to learn more about objective mismatch in RL.

The writing is clear and the flow is good. The notation of the summarized work has been standardized and made consistent across the whole survey.

I overall enjoyed reading this paper and only have superficial comments and corrections which I discuss below.

---

> ### Author Response · Authors · 2023-12-21
>
> We thank the reviewer for the detailed review of the paper and the valuable feedback. We have made corresponding changes in the paper highlighted in blue. Below, we address the reviewer's comments in a point-by-point manner.
>
> **Necessary correction 1**: Eq. (33) to (37) are missing "min" on the rhs of the equation.
>
> **Our response**: Thank your for highlighting this. We have changed the notation for all objective function/optimization problems by removing the loss function symbol $L_{method}(\cdot)$. We highlight these objective function by instead annotating the algorithm names in the equation numbers.
>
> **Necessary correction 2**: p.16, "Empirically, the authors showed that OMD [...]", this is ambiguous. Which authors? I presume this refers to Nikishin et al. but other work was cited since they were referenced last.
>
> **Our response**: Thank you for suggesting this. We have replaced "the authors" with "Nikishin et al.".
>
> **Recommendation 1**: Overall, I recommend making the distinction between optimal state/state-action value functions, state/state-action value functions and approximate state/state-action value functions explicit with separate notation, e.g., $Q^*, Q^{\pi}, \hat{Q}^*, \hat{Q}^{\pi}$.
>
> **Our response**: Thank you for the suggestion. We added a paragraph at the end of section 2.2 to clarify value function notations. Per other reviews' request, we have changed our notation of the true and learned dynamics models to $M$ and $\hat{M}$. We now use $Q_{M}^{\pi}(s, a)$ and $V_{M}^{\pi}(s)$ to refer to estimated value functions associated with a specific dynamics $M$ and policy $\pi$. When it is clear from context, we drop $M$ and $\pi$ to refer to the optimal policy w.r.t. learned dynamics, since all reviewed methods are model-based. We do not use extra notation to differentiate true value functions from estimated value functions because the true value functions can not be directly obtained and all value functions reviewed are estimates.
>
> **Recommendation 2**: The paper might benefit from a summarization of the motivations of the compared methods, possibly in a similar style as tables 1 and 2 or, alternatively, as a few paragraphs in the discussion section. For instance, some papers seem to have a focus on the offline RL setting and some more on model-misspecification. Summarizing this information would help guide readers to the papers most relevant to the motivation behind their current research. Though it's also possible that there isn't enough differences between the stated (or inferred) motivations of each paper to be worthwhile, but it would be a nice addition if there is.
>
> **Our response**: Thank you for the suggestion. We have added a sentence to section 5.5 on evaluation and benchmark to state that the top two motivator of decision-aware MBRL are model misspecification and model-exploitation which are backed by the theorem 2.1. We found this the most natural place to make this addition.
>
> **Recommendation 3**: Eq (13), In what sense is
> $\tilde{Q}$ a dual? Maybe add a sentence or two to clarify.
>
> **Our response**: We have revised this part to improve clarity. We have added descriptions that the dual value function $\tilde{Q}(s, a)$ is introduced to penalize the occupancy measure $d_{\hat{M}}^{\pi}$ from violating the Bellman flow constraint in the dual RL framework. The optiaml dual value function, which can be estimated from collected data, is used to compute the importance weights for weighted MLE model learning.
>
> **Recommendation 4**: Sec 4.2, in what way is
> $\mathcal{O}$ representing "optimality"? Is there any justification or explanation that could be summarized and included in this survey?
>
> **Our response**: We have added text before (20) to clarify this. The updated text is "Most control-as-inference methods define optimality
> using a binary variable $\mathcal{O}$ where the probability that an observed state-action pair is optimal ($\mathcal{O} = 1$) is
> defined as follows".
>
> **Recommendation 5**: bottom p. 11, "While theoretical understanding of self-predictive objectives is still lacking [...]", Subramanian, et al. [1] provide some theoretical discussion about self-predictive objectives and provide conditions under these latent variables (which they call information states) satisfy a form of value-equivalence.
>
> **Our response**: Thank you for suggesting this paper. We have added this to the respective section to highlight that the latent state $z$ learned by ALM via self-prediction is a sufficient statistic (or information state) for learning optimal policies w.r.t. the true environment without necessarily having to reconstruct environment observations.

---

> ### Author Response · Authors · 2023-12-21
>
> **Recommendation 6**: Control-as-inference summary, It's not clear to me what exactly is meant by hand-designed.
>
> **Our response**: We meant to say the factorization of the prior and variational distributions can involves a significant amount of human design and not automated. For example, if the encoder and latent dynamics model in (30) and (31), which were carefully designed by the authors, were replace by two different but equally valid (conditional) probability distributions, then various properties of the ALM algorithm may longer hold.
>
> **Recommendation 7**: Sec 4.3.2 felt generally too sparse on details and discussion. For instance, in the robust control summary, It's wasn't clear to me where the discussion about pessimism came from.
>
> **Our response**: Thank you for highlighting this. We now introduce the connection between robust control and value-prediction and the origin of pessimism more directly in the first paragraph of this section via "the robust control approaches reviewed in this section are in fact derived from the same value-equivalence principle" and "As we discuss below, these approaches highlight an inherent pessimism in value-equivalent dynamics models which is neglected in the value-prediction approaches". We have also rewritten the section on RAMBO significantly to highlight its connection with LAMPS.
>
> **Recommendation 8**: p. 16, the name OMD was taken from Bacon, et al. [2]. In addition, it would be good to discuss this work as they take a different approach to implicit differentiation and instead consider using the Lagrangian to solve the Bi-level optimization problem.
>
> **Our response**: Thank you for suggesting this paper. We have added it to the paragraph at the end of section 4.4 which discusses differentiable planning for imitation learning.
>
> **Recommendation 9**: p. 20, "we can classify distribution correction and differentiable planning as policy-based [...]", I'm not sure I would classify differentiable planning as being only policy-based or value-based. Some differentiable planning methods (e.g., OMD) learn a state-action value function.
>
> **Our response**: We believe this classification of differentiable planning is adequate under our definition of value-based methods, which are methods where explicit representation and estimation of value function is required (e.g., value prediction). This is similar to policy gradient being policy-based methods while most existing policy gradient algorithms also carry an explicit value function estimator.
>
> **Recommendation 10**: mp. 21, "decision-aware MBRL has focused on the single-task setting" \& "transfer learning presents a larger challenge for value optimization-equivalent agents", I would argue the focus of value iteration networks (Tamar et al., 2016) and QMDP-Net (Karkus et al., 2017) is multi-task/transfer learning. The motivation for these approach is to generalize over task specification (e.g., a floor map).
>
> **Our response**: Thank you pointing this out. We have revised the text to state QMDP-net and VIN's success on transfer learning in section 5.4. In addition, we have included and added discussions of 2 recent papers on TD-MPC, a value-prediction algorithm which has shown substantial performance improvement on multi-task training. These results also collectively suggest that the relationship between task transfer, decision-aware MRBL, and model identifiability is still an open question, which we discuss in section 5.4.

---

### Review · Reviewer_uYW8 · 2023-12-11

**Summary Of Contributions:**

This work is a survey to approaches related to addressing the objective-mismatch problem in model-based RL. They group the previous approaches into four broad categories, and provide an overview of each category and the high-level approaches used for each. They finally provide a lengthy discussion, and suggest a unified principle of value optimization-equivalence: the joint training of the model and the policy.

**Audience:**

Yes

**Broader Impact Concerns:**

The paper's broader impact statement seems sufficient.

**Claims And Evidence:**

Yes

**Requested Changes:**

I request the weaknesses highlighted above are addressed (I think they are mostly minor and will help readers understand the paper).

**Strengths And Weaknesses:**

# Strengths
- The classification of previous approaches into the four categories is very useful to gain an understanding of the field.
- The paper is well-written, and flows nicely between sections.
- Each approach discussed is explained in a sufficient level of depth.

# Weaknesses
- The notation $P$ is overloaded as both the transition operator of the MDP and the symbol for an underlying probability (e.g. $P(\tau)$, $P(a_t|s_t)$, $P(\mathcal{O}_t)=1$, among others). A bit more care with the notation would make it easier for people to understand.
- The term "homomorphic" appears in Section 4 with no prior definition: "Value-Equivalence searches for models that are homomorphic to the true environment and value function." Homomorphisms are a well-established concept in algebra (and more recently RL) but this seems to use neither definition. I would suggest the authors to either define this term as used in their work or use another term to avoid confusion.
- Almost all losses/objectives are not written properly mathematically. For example, equation (5) states $\max_{\hat{P}} L_{MLE}(\hat P) = \mathbb{E}_{s,a,s'}[\log(\hat{P}(s'|s,a) )] $ -- I know that the authors are trying to convey that we want to maximize the RHS, but as it is currently written it is not mathematically correct, and is likely to cause confusion (it at least caused confusion for me in the later sections).
- "Then the optimal policy...." This language suggests that the optimal policy is unique.
- Very minor: "f-divergence" => "$f$-divergence".
- "We refer to the process of finding the optimal policy for an MDP, including these variations, as policy optimization" - the authors can use any terms they please, however policy optimization is commonly reserved for policy-based methods in RL and using this terminology is likely to confuse readers who don't read the background closely.

---

> ### Author Response · Authors · 2023-12-21
>
> We thank the reviewer for the detailed review of the paper and the valuable feedback. We have made corresponding changes in the paper highlighted in blue. Below, we address the reviewer's comments in a point-by-point manner.
>
> **Weakness 1**: The notation $P$ is overloaded as both the transition operator of the MDP and the symbol for an underlying probability (e.g. $P(\tau), P(a_t|s_t), P(\tilde{O})$ among others). A bit more care with the notation would make it easier for people to understand.
>
> **Our response**: Thank you for highlighting this. We have changed notation of the true environment dynamics and learned dynamics to $M$ and $\hat{M}$, respectively.
>
> **Weakness 2**: The term "homomorphic'' appears in Section 4 with no prior definition: ``Value-Equivalence searches for models that are homomorphic to the true environment and value function.'' Homomorphisms are a well-established concept in algebra (and more recently RL) but this seems to use neither definition. I would suggest the authors to either define this term as used in their work or use another term to avoid confusion.
>
> **Our response**: Thank you for highlighting this. We have dropped the term "homomorphism" in the paper and use ``equivalence" instead to refer to equivalence in value estimation or optimization.
>
> **Weakness 3**: Almost all losses/objectives are not written properly mathematically. For example, equation (5) states $\max_{\hat{P}} L_{\text{MLE}}(\hat{P}) = \mathbb{E}_{(s, a, s') \sim \mathcal{D}}\left[\log \hat{P}(s'|s, a)\right]$ -- I know that the authors are trying to convey that we want to maximize the RHS, but as it is currently written it is not mathematically correct, and is likely to cause confusion (it at least caused confusion for me in the later sections).
>
> **Our response**: Thank your for highlighting this. We have changed the notation for all objective function/optimization problems by removing the loss function symbol $L_{method}(\cdot)$. We highlight these objective function by instead annotating the algorithm names in the equation numbers.
>
> **Weakness 4**: "Then the optimal policy...." This language suggests that the optimal policy is unique.
>
> **Our response**: Thank your for highlighting this. We found that the term ``the optimal policy" is frequently used in the RL literature and skipping the nuances about the optimal policy being not necessarily unique makes our writing more efficient without causing confusion.
>
> **Weakness 5**: Very minor: "f-divergence" => "$f$-divergence".
>
> **Our response**: Thank you. We have corrected this.
>
> **Weakness 6**: "We refer to the process of finding the optimal policy for an MDP, including these variations, as policy optimization" - the authors can use any terms they please, however policy optimization is commonly reserved for policy-based methods in RL and using this terminology is likely to confuse readers who don't read the background closely.
>
> **Our response**: We agree that the term "policy optimization" is usually reserved for policy-based methods. However, given the reviewed works have used a large variety of methods for optimizing policies, including both value-based, policy-based, and model-predictive control/trajectory optimization algorithms, we found it difficult to find another unifying term than "policy optimization". In order to avoid potential confusion, we have italicized the term ``policy optimization" in the paper.

---

### Author Response · Authors · 2023-12-21
**General response**

We would like to sincerely thank the reviewers for their thorough and insightful comments. Your feedback has helped us improve the manuscript substantially. Additionally, we believe that we have addressed all of the reviewers concerns. In the updated manuscript, we have highlighted all revisions in blue. We summarize the major revisions here and respond to individual comments below:

* We have added discussions on meta RL, Bayesian RL, and active inference in section 3. We have added discussions on value optimization-equivalence vs. value-equivalence in section 5.1, and multi-task RL in section 5.4.
* We included a new algorithm, TD-MPC [1], to the value-prediction family in section 4.3.1.
* We revised the robust control section (4.3.2) with additional context to improve the flow.
* We have removed the loss function symbol $L(\cdot) = $ from all objective functions. We have also added a paragraph at the end of section 2.2 to clarify value function notations with respect to different dynamics and policies. We now use $M$ to denote dynamics model and reduce the overloading of notation $P$.

[1] Hansen, N., Wang, X., & Su, H. (2022). Temporal difference learning for model predictive control. arXiv preprint arXiv:2203.04955.

---

### Decision · Action_Editor_VHoV · 2024-02-02

**Recommendation:** Accept as is

**Comment:**

This paper provides a comprehensive survey on an emerging sub-area of model-based reinforcement learning that concerns the problem of objective mismatch. The objective mismatch problem refers to the fact that many conventional model-based RL algorithms use different objectives for policy training (maximizing the return) and the model training (accurate prediction of the world, ignoring its role in the policy decision making). On the other hand, the more recent decision-aware model-based RL approaches learn a model that is aware of its role in the agent's decision making. This paper is a timely survey of this subarea of model-based RL.

The paper categorizes methods addressing this objective mismatch into four groups of Distribution Correction, Control-As-Inference, Value-Equivalence, and Differentiable Planning. It summarizes several papers in each group and discusses their relationship.

All reviewers are positive (three Accepts, one Weak Accept). They consider the paper a nice, timely, and comprehensive survey. The authors revised the paper based on the received feedback and it appears that the reviewers are happy about the changes. Therefore, I recommend the acceptance of the paper as is.

As a minor suggestion, the authors may consider citing the accepted paper of paper, instead of their arXiv version.

**Audience:**

This paper is very relevant to those interested in the model-based reinforcement learning.

**Claims And Evidence:**

The reviewers agree that the paper provides a nice and thorough survey of the subarea of objective mismatch in model-based reinforcement learning literature. The notation of the summarized papers have been made consistent across the paper.